# The DEG/ENaC cation channel protein UNC-8 drives activity-dependent synapse removal in remodeling GABAergic neurons

Tyne W Miller-Fleming[1†], Sarah C Petersen[2†‡], Laura Manning[3], Cristina Matthewman[4], Megan Gornet[2], Allison Beers[2], Sayaka Hori[5], Shohei Mitani[5], Laura Bianchi[4], Janet Richmond[3], David M Miller III[1,2*]

[1]Neuroscience Program, Vanderbilt University, Nashville, United States; [2]Department of Cell and Developmental Biology, Vanderbilt University, Nashville, United States; [3]Department of Biological Sciences, University of Illinois at Chicago, Chicago, United States; [4]Department of Physiology and Biophysics, University of Miami, Miami, United States; [5]Department of Physiology, Tokyo Women's Medical University, Tokyo, Japan

*For correspondence: david.miller@vanderbilt.edu

[†]These authors contributed equally to this work

Present address: [‡]Department of Neuroscience, Kenyon College, Gambier, United States

Competing interests: The authors declare that no competing interests exist.

**Abstract** Genetic programming and neural activity drive synaptic remodeling in developing neural circuits, but the molecular components that link these pathways are poorly understood. Here we show that the *C. elegans* Degenerin/Epithelial Sodium Channel (DEG/ENaC) protein, UNC-8, is transcriptionally controlled to function as a trigger in an activity-dependent mechanism that removes synapses in remodeling GABAergic neurons. UNC-8 cation channel activity promotes disassembly of presynaptic domains in DD type GABA neurons, but not in VD class GABA neurons where *unc-8* expression is blocked by the COUP/TF transcription factor, UNC-55. We propose that the depolarizing effect of UNC-8-dependent sodium import elevates intracellular calcium in a positive feedback loop involving the voltage-gated calcium channel UNC-2 and the calcium-activated phosphatase TAX-6/calcineurin to initiate a caspase-dependent mechanism that disassembles the presynaptic apparatus. Thus, UNC-8 serves as a link between genetic and activity-dependent pathways that function together to promote the elimination of GABA synapses in remodeling neurons.

## Introduction

Neural circuits are reorganized during development to produce a mature, functional nervous system. Circuit refinement involves both the elimination of specific synapses and the strengthening of others. These developmental changes in brain architecture are temporally coordinated with critical periods in which neuronal wiring can be shaped by activity. The establishment of binocular vision in the mammalian brain, for example, depends on visual input during a distinct developmental period in which the circuit is uniquely sensitive to neural activity. GABA signaling regulates the onset and duration of this critical period of plasticity in the visual circuit, but the molecular mechanisms that drive synaptic remodeling in this pathway are poorly defined (*Deidda et al., 2015*; *Hensch, 1998*).

Calcium influx by voltage-gated calcium channels (VGCCs) triggers neurotransmitter secretion in active neurons (*Catterall et al., 2008*). Elevated intracellular calcium can also regulate synaptic strength by controlling gene expression and protein function (*Flavell and Greenberg, 2008*; *Baumgärtel and Mansuy, 2012*). For example, the calcium-sensitive phosphatase, calcineurin,

**eLife digest** The brain contains billions of nerve cells, or neurons, that communicate with one another through connections called synapses. As the brain develops, these circuits are extensively modified as new synapses are created and others are removed. Neurological disorders may emerge if these processes are not regulated correctly. Identifying the biological pathways that control the addition and removal of synapses could therefore provide new insights into how to treat human brain diseases.

To communicate across a synapse, the signaling neuron releases chemicals called neurotransmitters that alter the activity of the receiving neuron. Some neurotransmitters, such as GABA, inhibit the activity of the receiving neuron. The activity of a neuron – and hence how often it releases neurotransmitters – depends on different ions moving into and out of the neuron through proteins called ion channels that are embedded in the cell membrane. For example, the movement of calcium ions into the neuron can trigger the release of neurotransmitters.

The roundworm *Caenorhabditis elegans* is often used as a model organism to study how the brain develops. During development, the worm nervous system eliminates synapses that release GABA and reassembles them at new locations. However, the nervous system does not eliminate these synapses at random. Miller-Fleming, Petersen et al. now show that a *C. elegans* protein called UNC-8 is responsible for this effect. UNC-8 forms part of an ion channel that allows sodium ions to enter the neuron and is selectively produced in GABA neurons that are destined for remodeling.

Miller-Fleming, Petersen et al. found that inside GABA-releasing neurons, calcium ions stimulate an enzyme called calcineurin that may in turn activate UNC-8. Sodium ions then enter the neuron through UNC-8 channels. This boosts the activity of the calcium ion channels, which further increases how many calcium ions enter the cell. Ultimately, the amount of calcium inside the neuron becomes high enough to activate an additional pathway that eliminates the synapse. This downstream pathway involves components of a cell-killing (or "apoptotic") mechanism that is repurposed in this case to remove the GABA release apparatus at the synapse.

Other proteins are likely to help UNC-8 sense the activity of neurons and destroy synapses in response. Further work is required to investigate these additional components and to determine how they work with UNC-8 to remove synapses in the nervous system during development.

antagonizes long term potentiation (LTP) through dephosphorylation of an AMPA receptor subunit that results in its endocytosis from the postsynaptic membrane (*Baumgärtel and Mansuy, 2012*; *Winder et al., 1998*). Recent work has shown that functional synaptic components can also be removed by the canonical apoptotic protease, caspase-3 (*Ertürk et al., 2014*; *Wang et al., 2014*). In *C. elegans*, the cell death pathway component CED-3/caspase-3 and its upstream regulator CED-4/Apaf1 promote synaptic disassembly by activating the F-actin severing protein gelsolin; the calcium sensitivity of this pathway points to a potential role for neuronal activity in synaptic remodeling (*Pinan-Lucarre et al., 2012*; *Meng et al., 2015*; *Liu et al., 2011*).

Degenerin/Epithelial Sodium Channels (DEG/ENaCs) are voltage-insensitive, trimeric cation channels. The term degenerin arises from the finding that constitutively active forms of DEG/ENaC channels induce neurodegeneration (*Bianchi and Driscoll, 2002*). ENaCs are expressed in mammalian epithelial tissues, where they promote sodium reabsorption (*Wemmie et al., 2002*). Emerging evidence indicates that DEG/ENaCs may also influence neuronal plasticity and synaptic function (*Wemmie et al., 2002*, *2003*; *Kreple et al., 2014*). For example, the acid-sensing DEG/ENaC protein, ASIC1a, is expressed in distinct brain regions where it promotes learning and memory (*Wemmie et al., 2003*, *2004*; *Zha et al., 2006*; *Ziemann et al., 2009*). Models to explain this role of DEG/ENaCs in synaptic plasticity have posited that depolarizing DEG/ENaC-dependent cation transport activity could enhance the activation of voltage-sensitive postsynaptic ion channels (*Wemmie et al., 2006*). DEG/ENaCs can also function in the presynaptic compartment to elevate neurotransmitter secretion (*Cho and Askwith, 2008*; *Voglis and Tavernarakis, 2008*; *Younger et al., 2013*). In this case, the depolarizing activity of a presynaptic ENaC channel is proposed to enhance neurotransmitter release by elevating calcium import at local

VGCCs (*Younger et al., 2013*). Here, we describe a DEG/ENaC protein that likely exerts a similar effect on intracellular calcium, but with the strikingly different downstream outcome of presynaptic destruction.

In the nematode *C. elegans*, a simple, well-defined GABAergic circuit is remodeled during development in a mechanism that is accelerated by neural activity (*White et al., 1978*; *Hallam and Jin, 1998*; *Park et al., 2011*; *Thompson-Peer et al., 2012*). Dorsal D (DD) motor neurons initially form synapses with ventral muscles (*Figure 1A*), which are later eliminated and relocated to dorsal muscles (*Figure 1B*). Ventral D (VD) motor neurons generate ventral synapses and express the transcriptional repressor protein UNC-55, which prevents these cells from remodeling; thus both DD and VD synapses are remodeled in *unc-55* mutants (*Figure 1B,C*) (*Walthall and Plunkett, 1995*; *Zhou and Walthall, 1998*; *Shan et al., 2005*). Several lines of evidence indicate that UNC-55 blocks the native DD remodeling program: (1) Ectopic expression of UNC-55 in DD neurons prevents synaptic remodeling (*Shan et al., 2005*); (2) Ventral VD synapses are initially established and then removed in *unc-55* mutants in a sequence that parallels the DD remodeling program (*Petersen et al., 2011*; *Meng et al., 2015*); (3) VD neurons form ectopic but functional synapses with dorsal muscles in *unc-55* mutants (*Thompson-Peer et al., 2012*); (4) The transcription factor genes *irx-1* and *hbl-1,* which UNC-55 negatively-regulates, normally function in DD neurons to promote remodeling (*Petersen et al., 2011*; *Thompson-Peer et al., 2012*). We exploited the anti-remodeling role of UNC-55 in a screen designed to detect key components of the DD remodeling program as RNAi hits that prevent the ectopic remodeling phenotype of *unc-55* mutants. This approach revealed a molecularly diverse array of genes that drive removal of GABA synapses (*Petersen et al., 2011*). Here we demonstrate that one of these UNC-55 targets, the DEG/ENaC protein UNC-8, functions to dismantle the presynaptic apparatus in remodeling GABA neurons (*Figure 1D,E*). The necessary roles of UNC-2/VGCC and TAX-6/Calcineurin in this synapse elimination mechanism argue that intracellular calcium is involved. Our results indicate that UNC-8 promotes synapse removal downstream of TAX-6 and thus could be potentially activated by TAX-6/Calcineurin. In turn, we suggest that the depolarizing activity of UNC-8 DEG/ENaC could enhance UNC-2/VGCC function. The net positive feedback loop involving UNC-2/VGCC, TAX-6/Calcineurin and UNC-8/DEG/ENaC could elevate presynaptic calcium above a critical threshold to activate a downstream caspase-dependent pathway leading to destabilization of the presynaptic apparatus (*Meng et al., 2015*). Thus, we propose a model in which UNC-8 triggers a synaptic removal mechanism that is regulated by the intersection of a genetic program that controls UNC-8 expression and a calcium-signaling pathway that promotes neurotransmission as well as synapse elimination.

## Results

### UNC-8 is expressed in remodeling D-class motor neurons

Our previous microarray data indicated that *unc-8* expression is elevated in GABA neurons in *unc-55* mutants (*Petersen et al., 2011*). As an independent test of this finding and to determine if *unc-8* is also expressed in DD neurons, we used a GFP reporter gene (*punc-8::GFP*) that includes a 1.6 kb genomic region upstream of the *unc-8* coding sequence (*Etchberger et al., 2007*). In wild-type adults, *punc-8::GFP* is expressed in DD GABAergic neurons, but shows little detectable GFP signal in VD motor neurons (*Figure 1F,G*). In an *unc-55* mutant, however, *punc-8::GFP* expression was clearly visible in both DD and VD neurons. As an additional test of this model, we fused GFP to the UNC-8 C-terminus in a fosmid reporter gene that spans the *unc-8* locus to include large flanking regions. The resultant UNC-8::GFP fusion protein is functional in vivo and is expressed in DD neurons in the wild type and also in VD neurons in an *unc-55* mutant. In this case, we also detected strong UNC-8::GFP expression in adjacent ventral cord DA and DB cholinergic neurons as predicted by the selective degeneration of these cells in a mutant that encodes a constitutively active UNC-8 protein (*Wang et al., 2013*). These results suggest that the fosmid reporter likely includes gene regulatory regions that direct expression of the native UNC-8 protein (*Figure 1—figure supplement 1A–C*). Together, these findings establish that UNC-8 expression is correlated with the execution of a genetically controlled program that promotes remodeling of GABAergic synapses.

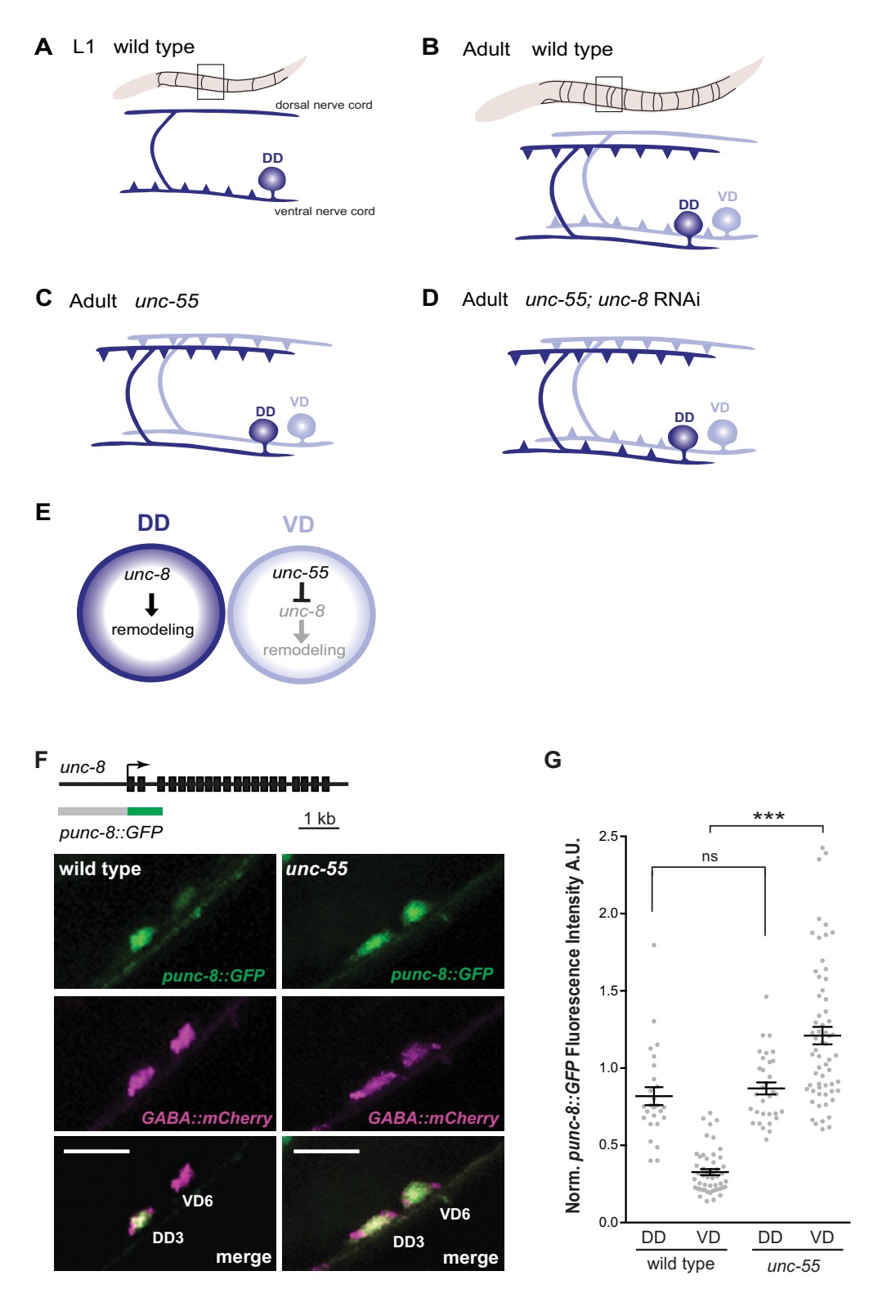

**Figure 1.** GABAergic neuron synaptic remodeling is transcriptionally controlled and depends on UNC-8. (**A**) Dorsal D (DD) GABAergic motor neurons (dark blue) synapse with ventral muscles during embryonic development. (**B**) DD synapses are relocated to dorsal muscles at the end of the first larval stage (L1). Ventral D (VD) GABAergic motor neurons (light blue) are generated in the late L1 and innervate ventral muscles. (**C**) The COUP/TF transcription factor UNC-55 is expressed in VD neurons and blocks remodeling; VD neurons relocate synapses to the dorsal side in loss-of-function *unc-*

*Figure 1 continued on next page*

*Figure 1 continued*

55 mutants. (**D/E**) UNC-8/DEG/ENaC expression is negatively regulated by UNC-55 and RNAi knockdown of *unc-8* suppresses ectopic VD remodeling in *unc-55* mutants. These results suggest that UNC-8 may also promote DD remodeling. Schematics modified from *Petersen et al. (2011)*. (**F**) *unc-8* expression in remodeling neurons is visualized with a p*unc-8::GFP* reporter gene. Strong p*unc-8::GFP* (green) expression was observed in DD motor neurons in wild type, but was also detected in VD motor neurons in *unc-55* animals. GABAergic motor neurons are labeled with p*ttr-39::mCherry* (magenta). Scale bar is 10 μm. (**G**) Normalized fluorescence intensity is plotted on the Y-axis in arbitrary units (A.U.). p*unc-8::GFP* expression is enhanced in VDs, but not DDs, in *unc-55* mutants (***p<0.001, ns is not significant, One-Way ANOVA with Bonferroni correction). $n \geq 26$ DDs and $n \geq$ 51 VDs per genotype, data are mean ± SEM.

The following figure supplement is available for figure 1:

**Figure supplement 1.** UNC-8 is expressed in remodeling GABA neurons.

## UNC-8 promotes the removal of ventral DD synapses

Our RNAi results suggested that UNC-8 promotes synapse removal in remodeling GABA neurons (*Figure 1D,E*) (*Petersen et al., 2011*). To confirm this finding, we generated a loss-of-function allele *unc-8(tm5052)* that deletes 197 nucleotides of the first transmembrane domain and shifts the reading frame to introduce a premature stop codon (*Figure 2A*). In this configuration, *unc-8(tm5052)* should result in an unstable *unc-8* mRNA and likely null allele.

To ask if UNC-8 promotes DD synapse removal, we utilized a DD-specific promoter (p*flp-13*) to drive expression of the presynaptic vesicle-associated proteins, GFP::RAB-3 and SNB-1::GFP/synaptobrevin. DD remodeling is initiated toward the end of the first larval stage (L1) and involves the relocation of ventral presynaptic components to the dorsal side (*Figure 2B*, *Figure 2—figure supplement 1A*) (*Hallam and Jin, 1998*; *Petersen et al., 2011*). Dorsal and ventral synapses were quantified throughout the remodeling period in early stage larval animals by monitoring GFP::RAB-3 puncta. This experiment revealed that the removal of ventral DD synapses was significantly delayed in *unc-8* mutants compared to wild type (*Figure 2C*). This effect is unlikely due to slower overall development as the birth of post-embryonic VD neurons is unaffected in *unc-8* mutants (*Figure 2—figure supplement 1C*). In addition, residual ventral SNB-1::GFP puncta are visible in the DD neurons of adult *unc-8* animals, but rarely in wild type indicating that *unc-8* is required for the complete removal of ventral GABA synapses in remodeling DD neurons (*Figure 2D,E*). In contrast, assembly of dorsal DD synapses was not perturbed in *unc-8* animals (*Figure 2E*, *Figure 2—figure supplement 1B,D*). These results suggest that *unc-8* is required for the efficient removal of presynaptic components in the DD remodeling program, but is not necessary for assembly of nascent DD synapses with dorsal muscles (*Figure 2G*). The idea that *unc-8* promotes removal of ventral presynaptic components in the DD remodeling program is consistent with our finding that expression of a GFP-tagged UNC-8 protein in GABA neurons results in UNC-8::GFP puncta that are exclusively localized to the ventral nerve cord (*Figure 2F*).

## UNC-8 promotes synapse elimination in remodeling GABAergic neurons

Having shown that *unc-8* is required for the timely and efficient removal of ventral synapses in remodeling DD neurons, we next used *unc-8(tm5052)* to confirm this function in VD neurons that remodel in *unc-55* mutants. This test is important because previous evidence suggests that VD remodeling in *unc-55* mutants is driven by the ectopic activation of the native DD remodeling program (*Shan et al., 2005*; *Petersen et al., 2011*). Expression of the GABAergic presynaptic marker SNB-1::GFP in wild-type adults shows a uniform pattern of SNB-1::GFP puncta in the ventral nerve cord that corresponds to VD synapses with ventral muscles (*Figure 3A*) (*Hallam and Jin, 1998*). As previously reported, ventral SNB-1::GFP clusters are largely depleted in *unc-55* mutant animals due to ectopic VD remodeling (*Petersen et al., 2011*). We constructed an *unc-55; unc-8* double mutant to ask if *unc-8* is required for the removal of ventral synapses in remodeling GABA neurons. Both fluorescence intensity measurements as well as quantification of SNB-1::GFP puncta number revealed significant retention of the SNB-1::GFP signal in the ventral nerve cord of *unc-55; unc-8* mutant animals in comparison to *unc-55* alone (*Figure 3A–D*, *Figure 3—figure supplement 1A*). We also performed time course experiments to confirm that the retention of ventral GABA synapses during larval development in *unc-55; unc-8* animals persists into adulthood (*Figure 3—figure*

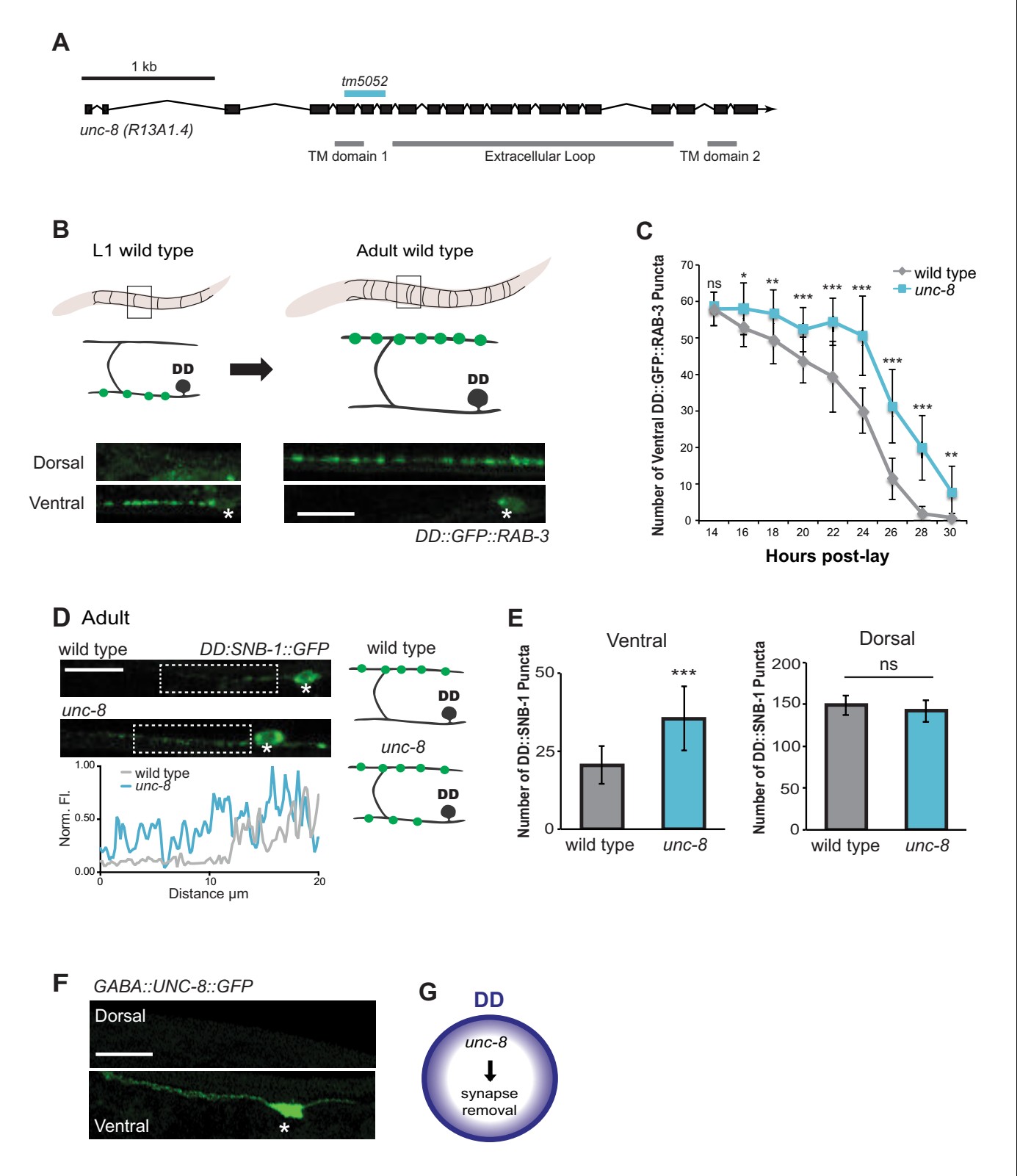

**Figure 2.** The DEG/ENaC subunit UNC-8 promotes removal of ventral DD synapses. (**A**) Schematic of the *unc-8* gene and predicted UNC-8 protein. DEG/ENaC channel subunits contain two transmembrane domains (TM domains) and a large extracellular loop (gray bars). The *unc-8* deletion allele *tm5052* is indicated (blue bar). (**B**) DD GABA neuron synapses (green) with ventral muscles are relocated to the dorsal side during development. DD-specific GFP-tagged RAB-3 (*pflp-13::GFP::RAB-3*) labels synapses in the ventral nerve cord of early L1 larvae and the dorsal nerve cord of adults.
*Figure 2 continued on next page*

*Figure 2 continued*

Asterisk denotes DD4 soma. Scale bar is 10 µm. (C) DD remodeling was quantified by counting GFP::RAB-3 puncta during larval stages. Removal of ventral DD synapses is significantly delayed in *unc-8* animals (*p<0.05, **p<0.01, ***p<0.001, ns is not significant, Student's *t*-test, data are mean ± SD). (D) Representative images of wild-type and *unc-8* adult ventral nerve cords (asterisk denotes DD5 soma). Ventral DD synapses labeled with GFP-tagged synaptobrevin (*pflp-13::SNB-1::GFP*) are retained in *unc-8* mutant adult animals. Scale bar is 10 µm. Inset shows pixel intensity over a 20 µm region (indicated by dashed boxes) of the ventral nerve cord in wild-type and *unc-8* animals. (E) Removal of *DD::SNB-1::GFP* puncta is defective in *unc-8* mutant adults; however, dorsal DD synaptic assembly is not affected (***p<0.001, $n \geq 20$, ns is not significant, Student's *t*-test, data are mean ± SD). (F) Ventral localization of GFP-tagged UNC-8 in GABA neurons. Asterisk denotes DD5 soma. Scale bar is 10 µm. (G) UNC-8 promotes removal of ventral presynaptic components in DD neurons, but is not required for dorsal synapse formation.

The following figure supplement is available for figure 2:

**Figure supplement 1.** UNC-8 removes ventral synapses, but is not required for assembly of dorsal synapses.

*supplement 1B*). Because this assay should detect ventral SNB-1::GFP puncta in both DD and VD neurons, we co-labeled these synapses with the DD-specific marker, *flp-13::mCherry::RAB-3* to identify the fraction of each GABA neuron type. This experiment revealed that approximately 40% of residual GABA synapses in *unc-55; unc*-8 animals are derived from DD neurons and the balance (~60%) from VD neurons (*Figure 3—figure supplement 1C*). Our results indicate that the wild-type *unc-8* gene is capable of promoting the removal of ventral synapses in both classes of GABA neurons thus supporting the hypothesis that the DD and VD remodeling programs utilize shared sets of active components (*Shan et al., 2005*; *Petersen et al., 2011*).

## UNC-8 expression in GABAergic neurons is required for synapse removal

Having established that UNC-8 is necessary for the complete removal of GABAergic presynaptic proteins (*Figures 2,3*) and that *punc-8::*GFP is expressed in remodeling DD and VD neurons (*Figure 1F*), we next asked if UNC-8 function is cell-autonomous. To address this question, we used cell-specific RNA interference (csRNAi) to knock down *unc-8* expression in GABA neurons. *unc-55* animals expressing empty vector RNAi display a stereotypical loss of ventral GABA synapses as predicted. In contrast, *unc-55* mutant VD neurons that express the *unc-8(csRNAi)* transgene show significant retention of GABAergic synapses in the ventral nerve cord (*Figure 3E,F*), thus demonstrating that *unc-8* functions in GABA neurons to promote the elimination of ventral synapses. As an additional test of this model, we restored wild-type *unc-8* function to *unc-55;unc-8* mutant animals by expressing *unc-8* cDNA in GABA neurons. In this case, SNB-1::GFP puncta are efficiently removed from the ventral nerve cord, thus confirming that *unc-8* exerts a cell-autonomous role in the disassembly of ventral GABAergic synapses (*Figure 3—figure supplement 2A*). Finally, we determined that forced expression of *unc-8* cDNA in wild-type VD neurons induces synapse elimination whereas the ventral synapses of neighboring VD neurons not expressing the UNC-8 transgene are maintained. In addition to confirming the active role of UNC-8 in GABA neuron synapse elimination, this experiment also demonstrates that expression of UNC-8 alone is sufficient to promote synaptic disassembly in a GABA neuron that otherwise maintains a stable, functional presynaptic apparatus (*Figure 3—figure supplement 2B*).

## UNC-8 promotes disassembly of the presynaptic complex in remodeling GABAergic neurons

Having confirmed that wild-type *unc-8* promotes the removal of the presynaptic proteins SNB-1::GFP and GFP::RAB-3 from the ventral nerve cord in remodeling GABA neurons (*Figure 2C–E*, *Figure 3*, *Figure 3—figure supplement 1*), we next used additional fluorescent markers to ask if the *unc-8* gene also drives the elimination of other key components of the presynaptic apparatus. For this experiment, we monitored fluorescent markers for the presynaptic density protein α-liprin/SYD-2 (*Dai et al., 2006*) and the membrane-associated endocytic protein endophillin/UNC-57 (*Schuske et al., 2003*). As observed for SNB-1::GFP, all the additional presynaptic proteins examined appear as distinct puncta that localize to the presynaptic membrane in wild-type and *unc-8* GABA neurons. This ventral localization is substantially reduced in *unc-55* animals. The partial

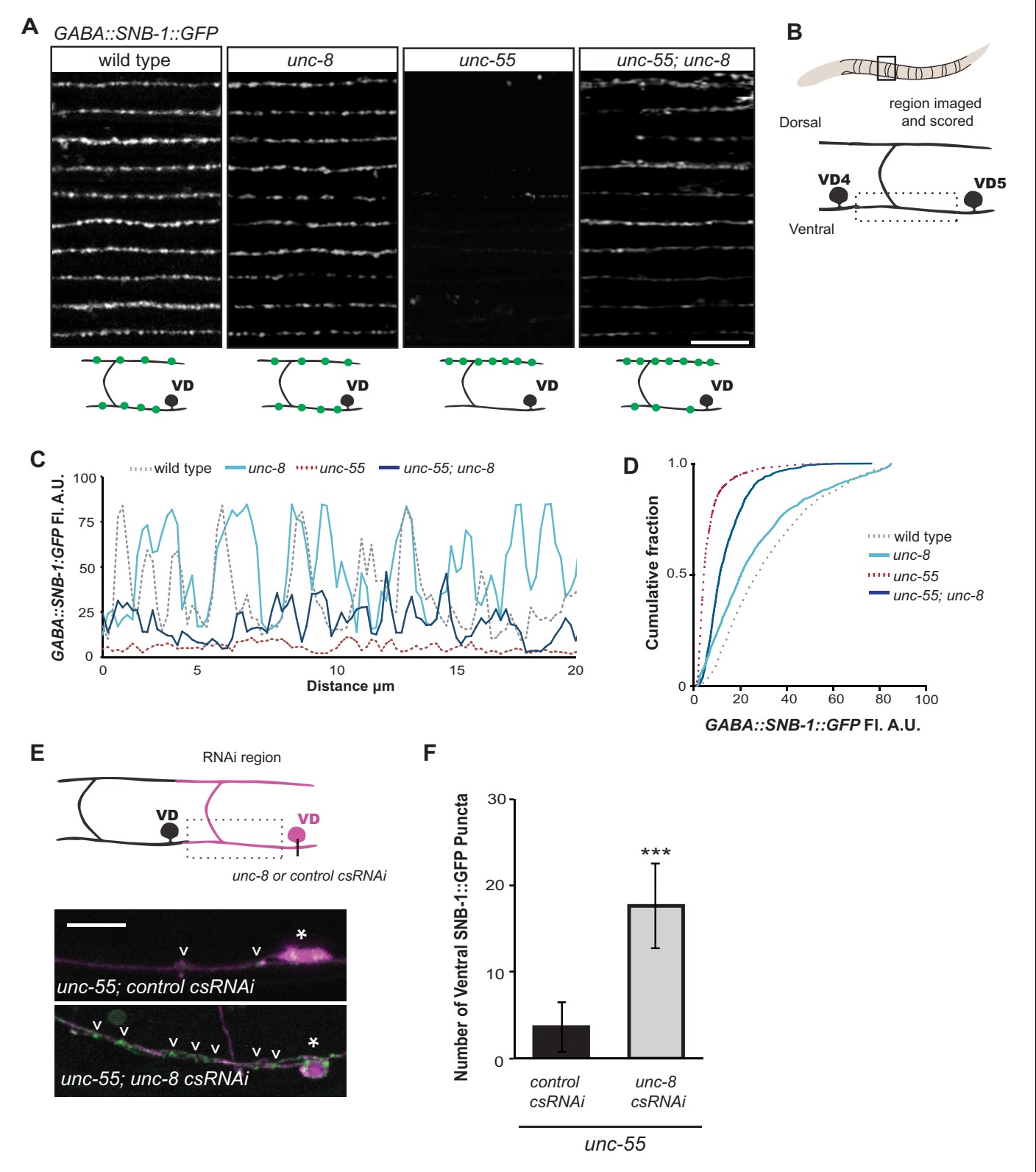

**Figure 3.** UNC-8 drives removal of ventral GABAergic synapses. (**A**) Ventral GABA synapses labeled with GFP-tagged synaptobrevin (*punc-25::SNB-1::GFP*) for 10 adult animals. Wild-type and *unc-8(tm5052)* show similar distributions of SNB-1::GFP puncta. Ventral SNB-1::GFP is depleted from *unc-55* due to VD remodeling, but partially restored in *unc-55; unc-8* animals. (**B**) Data for Figure panels **A**, **C**, **D** were collected from the ventral nerve cord between VD4 and VD5. (**C**) SNB-1::GFP fluorescent intensity measurements from each genotype. Each line represents the pixel intensity over a 20 μm

*Figure 3 continued on next page*

*Figure 3 continued*

region of the VNC from a single representative animal. (**D**) Cumulative frequency curves for SNB-1::GFP fluorescence intensity for each genotype (*n* > 10 animals). *unc-55* animals show a significant loss of ventral SNB-1::GFP fluorescence (p< 0.0001 vs wild type). SNB-1::GFP fluorescence is partially restored in *unc-55; unc-8* animals, demonstrating the role of UNC-8 for synapse removal (p<0.0001 vs *unc-55*). p values calculated with Kruskal-Wallis and Dunn's post test. (**E**) Knockdown of *unc-8* by GABA-neuron-specific RNAi (*unc-8 csRNAi*) restored SNB-1::GFP puncta to the VNC of *unc-55; juIs1* animals vs control animals expressing the empty vector RNAi (*control csRNAi*). GABAergic neurons are labeled with *punc-25::mCherry* (magenta). Asterisks denote GABA neuron cell bodies and arrowheads point to SNB-1::GFP-labeled ventral synapses. (**F**) Quantification of ventral synapses in the region anterior to each cell body expressing the RNAi construct (*n* ≥ 60 animals). RNAi knockdown of *unc-8* in *unc-55* mutant GABA neurons significantly suppresses synapse removal (***p<0.001, Student's *t*-test. Data are mean ± SD). Scale bars are 10 μm.

The following figure supplements are available for figure 3:

**Figure supplement 1.** UNC-8 promotes synapse disassembly in remodeling GABAergic neurons.

**Figure supplement 2.** UNC-8 functions cell autonomously and is sufficient to promote ventral synapse elimination in GABA neurons.

recovery of fluorescent puncta in *unc-55; unc-8* animals relative to *unc-55* shows that UNC-8 promotes the elimination of ventral SNB-1::GFP, UNC-57::GFP, mCherry::RAB-3, and SYD-2::GFP puncta (*Figure 3—figure supplement 1A*, *Figure 4A–E*). Because these components mark synaptic vesicles (SNB-1, RAB-3) as well as the presynaptic membrane (SYD-2, UNC-57), their removal predicts that *unc-8* could be required for dismantling the overall presynaptic apparatus. To investigate this possibility, we simultaneously imaged mCherry::RAB-3 and UNC-57::GFP to monitor the organization of the presynaptic domain in *unc-55; unc-8* animals. RAB-3 cycles on and off synaptic vesicles in a GTP/GDP-dependent manner and UNC-57/endophilin recycles between vesicles and the presynaptic membrane (*Schuske et al., 2003*; *Nonet et al., 1997*). As expected, mCherry::RAB-3 and UNC-57::GFP are strongly co-localized at ventral synapses in wild-type animals ($r^2 = 0.72 \pm 0.03$) (*Figure 4F*). Although fewer presynaptic clusters are detected at ventral synapses in *unc-55; unc-8* mutants, residual mCherry::RAB-3 and UNC-57::GFP puncta are comparably co-localized ($r^2 = 0.67 \pm 0.04$) (*Figure 4G*, *Figure 4—figure supplement 1A–C*).

We used electron microscopy (EM) to test this idea by examining the structure of the GABAergic synapses in *unc-55; unc-8*. At the EM level, both GABAergic and cholinergic synapses show electron-dense presynaptic terminals and abundant synaptic vesicles, but can be distinguished by morphological criteria (see Materials and methods). We identified ventral GABAergic synapses in wild-type, *unc-8* and in *unc-55; unc-8* animals (wild type = 6 synapses/20.08 μm, n = 3 animals, *unc-8* = 5 synapses/12.88 μm, n = 2 animals, *unc-55; unc-8* = 2 synapses/12.16 μm, n = 2 animals, *Figure 4H–J*, *Figure 4—figure supplement 1D*, and *Figure 4—figure supplement 2*). As expected, no ventral GABAergic synapses were detected in *unc-55* mutants (0 synapses/9.84 μm, n = 2 animals); whereas ventral cholinergic motor neuron synapses were preserved (data not shown). The finding that GABAergic synapses were detectable in *unc-55;unc-8* double mutants is consistent with the partial restoration of fluorescently-labeled presynaptic markers to ventral GABA neuron processes in *unc-55; unc-8* animals (*Figure 3A*, *Figure 3—figure supplement 1*, *Figure 4 A–G*) and therefore argues that the wild-type *unc-8* gene promotes disassembly of multiple components of the presynaptic apparatus.

Having confirmed that *unc-55; unc-8* animals contain GABAergic presynaptic domains we next used electrophysiological recordings to ask if these residual synapses are functional. Spontaneous inhibitory postsynaptic events (iPSCs) arising from GABA release were recorded from ventral body muscles (see Materials and methods). Robust, iPSCs were detected for wild-type and *unc-8* animals (*Figure 4K,L*). Ventral iPSCs were not produced in *unc-55* mutants as expected since these animals lack ventral GABAergic synapses (*Figure 3A*) (*Petersen et al., 2011*). Interestingly, ventral iPSCs were not restored in *unc-55; unc-8* animals despite the presence of organized clusters of fluorescent presynaptic proteins (*Figure 3A*, *Figure 4A–G*) and electron dense active zones with abundant synaptic vesicles (*Figure 4I–J*). Total ventral synaptic activity (cholinergic and GABAergic) was measured to confirm that spontaneous ventral cholinergic activity was unaffected by the *unc-55* and *unc-8* mutations (*Figure 4—figure supplement 1E–F*). We see no significant change in the expression or clustering of postsynaptic GABA$_A$ receptors (UNC-49::GFP) in *unc-55; unc-8* animals in comparison

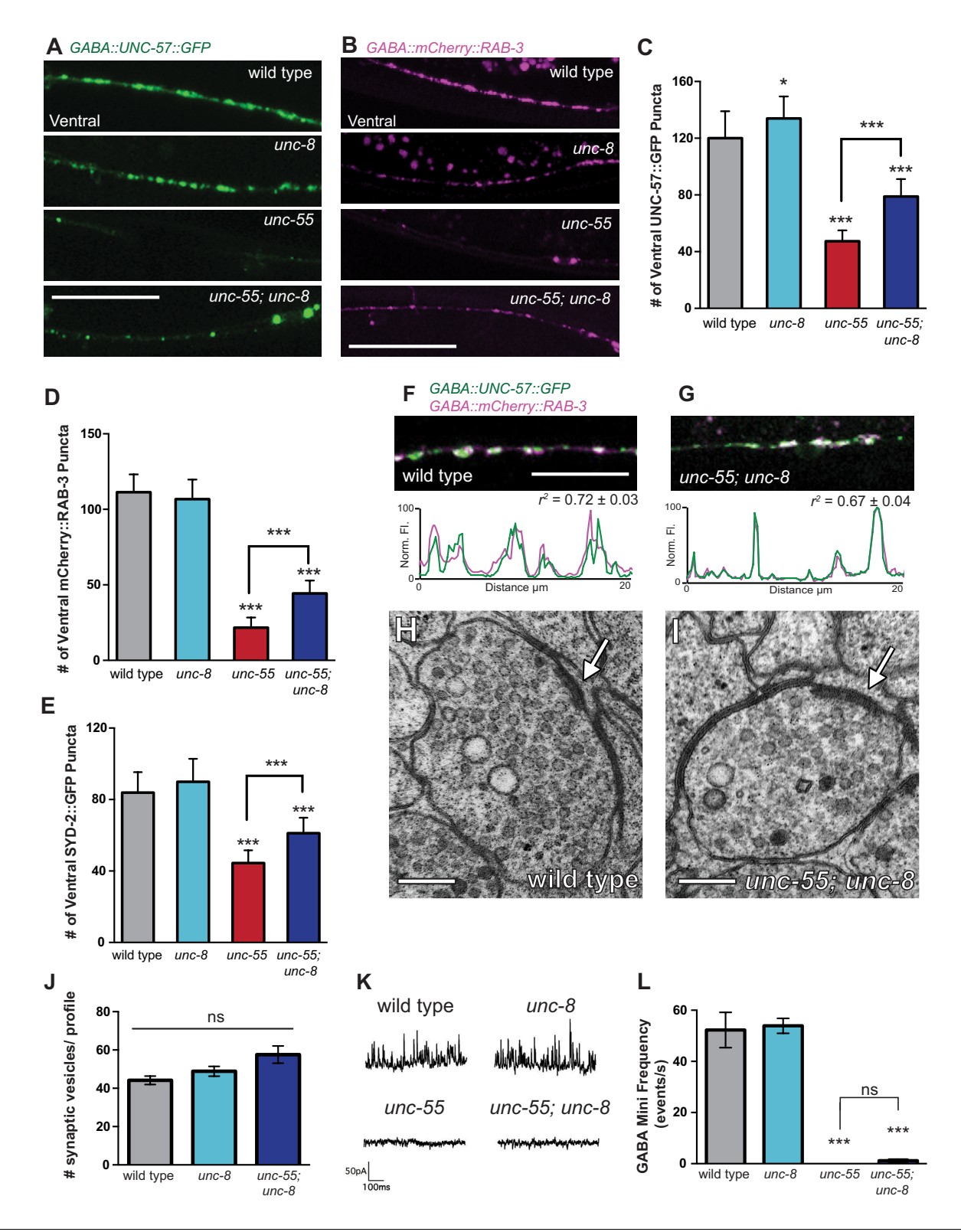

**Figure 4.** UNC-8 promotes disassembly of the presynaptic apparatus in GABAergic motor neurons. Fluorescent puncta for presynaptic proteins (UNC-57::GFP, mCherry::RAB-3, and SYD-2::GFP) were counted in the ventral nerve cord from VD3 to VD11. (A/C) Representative images (A) and quantification (C) of endophilin/UNC-57 indicate that *unc-8* promotes removal of UNC-57::GFP from ventral synapses in remodeling neurons (*p<0.05, ***p<0.001, *n* ≥ 25). (B/D) Representative images (B) and quantification (D) show reduced removal of the presynaptic G protein RAB-3 in *unc-55; unc-8*

*Figure 4 continued*

animals (***p<0.001, $n \geq 21$). Scale bars are 10 μM. (E) Efficient removal of the presynaptic density protein α-liprin/SYD-2 from ventral synapses in *unc-55* requires *unc-8* (***p<0.001, $n \geq 21$, One-Way ANOVA with Bonferroni correction, data are mean ± SD). (F/G) GFP-tagged endophilin (*punc-25::UNC-57::GFP*) and mCherry::RAB-3 (*punc-25::mCherry::RAB-3*) are co-localized in GABA neurons of wild-type and *unc-55;unc-8* animals. Representative images and normalized fluorescence intensity plots for a 20 μm region of the ventral nerve cord are shown. Scale bar is 10 μm. $r^2$ is Pearson's correlation coefficient ($n \geq 10$, mean ± SEM). Presynaptic components are co-localized in wild type ($r^2 = 0.72 \pm 0.03$) and *unc-55;unc-8* ($r^2 = 0.67 \pm 0.04$). Average $r^2$ value for *unc-55;unc-8* is not statistically different from the average $r^2$ value for wild type (p<0.001, Mann-Whitney test, see ***Figure 4—figure supplement 1***). (H/I) Electron micrographs of GABA synapses with ventral muscles in (H) wild type and (I) *unc-55;unc-8*. No ventral GABA presynaptic densities were detected in *unc-55*. Arrows point to presynaptic density, scale bars are 200 nm. (J) Synaptic vesicles were quantified in ventral GABAergic synapses. Synapses in wild-type, *unc-8* and *unc-55;unc-8* animals contain comparable numbers of synaptic vesicles (N > 5 for each genotype, ns is not significant). (K) Representative traces of ventral mini-iPSCs from each genotype. (L) The high frequency of ventral mini-iPSCs in wild-type and in *unc-8* animals were not observed in *unc-55* or *unc-55;unc-8* (***p<0.001, ns is not significant, $n \geq 5$, data are mean ± SEM, One-Way ANOVA with Bonferroni correction).

The following figure supplements are available for figure 4:

**Figure supplement 1.** Ventral synapses in *unc-55;unc-8* mutants are well-organized.

**Figure supplement 2.** GABAergic and cholinergic synapses are detectable in electron micrographs of wild-type and *unc-55; unc-8* animals.

**Figure supplement 3.** The postsynaptic UNC-49 GABA$_A$ receptor co-localizes with the presynaptic domains of remodeling GABAergic neurons.

to wild type; UNC-49::GFP is closely co-localized with presynaptic mCherry::RAB-3 at ventral *unc-8* and *unc-55; unc-8* GABAergic synapses both before remodeling and in adults (***Figure 4—figure supplement 3***) (***Gally and Bessereau, 2003***; ***Petersen et al., 2011***). Together, these data argue against the possibility that the absence of IPSCs in *unc-55; unc-8* animals is caused by a defective postsynaptic response to GABA and therefore favor the hypothesis that presynaptic dysfunction limits GABA secretion.

To summarize, we have shown that *unc-55* animals lack GABAergic neuromuscular junctions and that presynaptic components are partially restored in *unc-55; unc-8* mutants. Despite the presence of both presynaptic domains with abundant synaptic vesicles and clusters of postsynaptic GABA$_A$ receptors, residual GABAergic synapses in *unc-55; unc-8* animals are not functional. Taken together, these results indicate that UNC-8 is required for the efficient removal of the ventral presynaptic apparatus, but that the presynaptic defect of *unc-55; unc-8* animals could be due to an additional UNC-8-independent pathway, potentially involving the previously described homeodomain transcription factor *irx-1* (***Petersen et al., 2011***), acting in parallel to promote the disassembly of the ventral synaptic architecture of remodeling GABAergic neurons.

## UNC-8 channel activity is required for synapse removal

We have shown that UNC-8 protein preferentially gates sodium when expressed in *Xenopus* oocytes (***Wang et al., 2013***). We therefore considered the hypothesis that UNC-8 channel function is required for synaptic elimination. This idea is consistent with our finding that the *unc-8* mutant disrupts DD and VD synapse removal (***Figures 2C*** and ***3A***). As a specific test of this hypothesis, we treated *unc-55* mutant animals with Benzamil, which has previously been shown to inhibit DEG/ENaC channel activity (***Kleyman and Cragoe, 1988***). In this experiment, *unc-55* mutant animals were grown on media containing 3 mM Benzamil (see Materials and methods). Ventral SNB-1::GFP puncta were scored in adult animals. As expected, *unc-55* animals from control plates show few ventral SNB-1::GFP puncta in GABA neurons (***Figure 5A,B***). In contrast, treatment of *unc-55* animals with Benzamil results in a significantly larger number of residual SNB-1::GFP puncta on the ventral side (***Figure 5A,B***). However, Benzamil treatment of *unc-55;unc-8* animals does not induce any additional enhancement of ventral SNB-1::GFP puncta. This result argues that Benzamil antagonizes synaptic removal by specifically inhibiting UNC-8 (***Figure 5—figure supplement 1A***). To confirm that Benzamil inhibits UNC-8 channel activity, we expressed the constitutively active UNC-8(G387E) protein in *Xenopus* oocytes for electrophysiological analysis (***Figure 5—figure supplement 1B***). We have previously established that the UNC-8(G387E) protein shows robust sodium transport activity in an

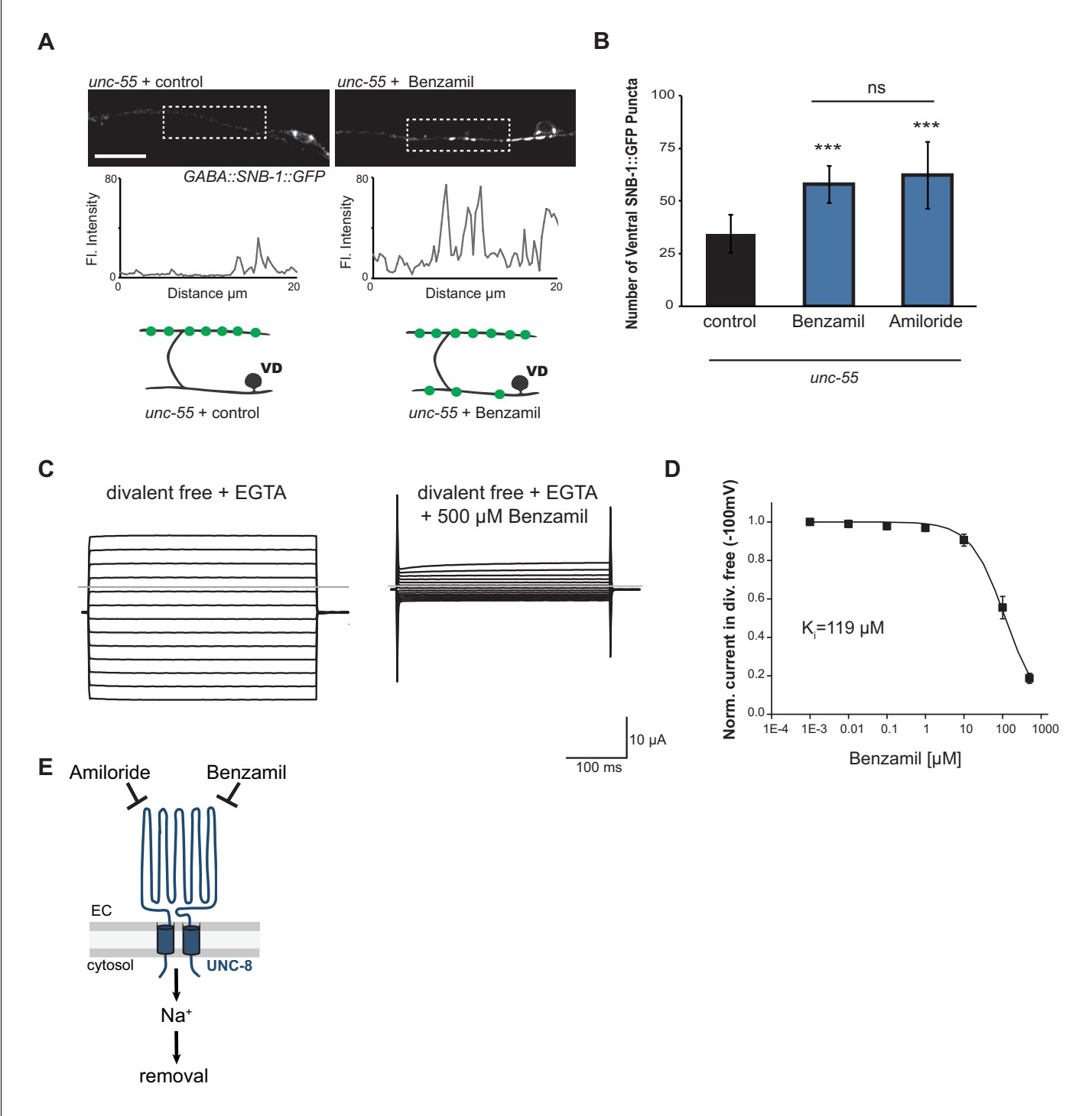

**Figure 5.** UNC-8 cation channel activity promotes the removal of ventral synapses in remodeling GABA neurons. (**A**) Representative images and fluorescence intensity plots (generated from 20 µm dashed region) of SNB-1::GFP-marked ventral GABA neuron synapses in *unc-55* animals treated with either 3 mM Benzamil or water (control). Scale bar is 10 µm. (**B**) Benzamil and Amiloride antagonize the removal of ventral GABA synapses in *unc-55* mutant animals. Ventral GABA neuron synapses were quantified by counting SNB-1::GFP puncta (***p<0.001, ns is not significant, $n \geq 25$ animals, data are mean ± SD, One-Way ANOVA with Bonferroni correction). (**C**) Benzamil blocks UNC-8(G387E) current in *Xenopus* oocytes. Representative currents from oocyte expressing UNC-8(G387E) in a bath of divalent cation-free solution plus EGTA. Currents elicited by 20mV voltage steps from -160mV to +100mV. The holding potential was -30mV. The gray line represents the zero current level (left). The same oocyte exposed to 500µM Benzamil (right). (**D**) Benzamil dose-response curve in divalent cation-free bath solution. Currents recorded with Benzamil were normalized against recordings in divalent cation-free bath solution plus EGTA at -100mV. Data were fitted to the Boltzmann's equation to derive $K_i$ = 119 µM ($n = 10$

*Figure 5 continued on next page*

*Figure 5 continued*

oocytes). Data are mean ± SEM. (**E**) Pharmacological inhibition of UNC-8 channel activity with either Benzamil or Amiloride blocks removal of ventral synapses in remodeling GABA neurons synaptic remodeling (EC is extracellular).

The following figure supplement is available for figure 5:

**Figure supplement 1.** UNC-8 is required for the inhibitory effect of Benzamil and Amiloride on synaptic removal.

extracellular solution that is depleted of calcium and other divalent cations (*Figure 5C*, *Figure 5—figure supplement 1C*) (*Wang et al., 2013*). Under these conditions, Benzamil strongly inhibits UNC-8(G387E) sodium transport with a $K_i \sim 119\ \mu M$ (*Figure 5C,D*). In the presence of extracellular calcium, Benzamil acts as a more potent inhibitor ($K_i = 47\ \mu M$) suggesting that calcium ions enhance Benzamil binding (*Figure 5—figure supplement 1C*). A similar effect of extracellular divalent cations on inhibitor potency was previously observed for the DEG/ENaC inhibitor Amiloride (*Wang et al., 2013*). Treatment with Amiloride also antagonizes the removal of ventral SNB-1::GFP puncta in *unc-55* mutant neurons (*Figure 5B*), but does not alter ventral SNB-1::GFP puncta in either *unc-8* or *unc-55;unc-8* animals (*Figure 5—figure supplement 1A*). Taken together, these results show that Amiloride and Benzamil disrupt the GABA neuron synaptic removal and are consistent with the idea that this effect is due to the inhibition of sodium transport by an UNC-8-containing DEG/ENaC channel (*Figure 5E*).

## UNC-8 removes synapses in an activity-dependent pathway

Previous work has shown that neuronal activity promotes DD presynaptic remodeling; the dorsal DD synapse formation is accelerated in mutants that enhance synaptic activity and delayed by genetic defects that block neurotransmission (*Thompson-Peer et al., 2012*). We predicted that neuronal activity is also required for the removal of ventral synapses in remodeling GABA neurons. Voltage-gated calcium channels (VGCCs) promote synaptic vesicle fusion and neurotransmitter release by elevating intracellular calcium (*Catterall et al., 2008*). In *C. elegans*, the *unc-2* gene encodes a CaV2 α1 subunit of a P/Q-type calcium channel (*Mathews et al., 2003*). UNC-2 is localized to the presynaptic terminals of ventral cord motor neurons and is required for neurotransmitter release (*Gracheva et al., 2008*; *Saheki and Bargmann, 2009*). We determined that *unc-55; unc-2* double mutants show a significantly greater number of SNB-1::GFP puncta on the ventral side in comparison to *unc-55* animals (*Figure 6A,B*). This result indicates that wild-type *unc-2* promotes synapse removal in remodeling GABA neurons. Because comparable levels of ventral SNB-1::GFP signal were observed in *unc-55; unc-8* animals, we next conducted a genetic test to ask if *unc-2* and *unc-8* eliminate ventral synapses by either independent or shared mechanisms. We observed no additional increase in the number of ventral cord SNB-1::GFP puncta in *unc-55; unc-8; unc-2* animals compared to either double mutant (*i.e., unc-55;unc-8* or *unc-55;unc-2*). This result favors a model in which *unc-2* and *unc-8* act together in a linear pathway to remove ventral synapses in remodeling GABA neurons (*Figure 6A,B,E*).

We performed an additional genetic experiment to confirm that UNC-8 functions in an activity-dependent pathway in remodeling DD motor neurons. The tomosyn protein, TOM-1, normally down-regulates neurotransmitter release by inhibiting synaptic vesicle (SV) priming (*Gracheva et al., 2006*; *McEwen, 2006*). We confirmed that a loss-of-function *tom-1/tomosyn* mutant, which enhances evoked SV fusion, results in precocious assembly of dorsal DD synapses (*Figure 6C*) (*Thompson-Peer et al., 2012*). This effect was not observed, however, in *tom-1;unc-8* double mutants which suggests that wild-type *unc-8* function is necessary for the accelerated remodeling of DD neurons in *tom-1* mutant animals (*Figure 6C*). Although we previously determined that UNC-8 is exclusively involved in the removal of ventral DD synapses in wild-type animals, this result suggests that UNC-8 is also necessary for the precocious assembly of dorsal DD synapses in *tom-1* mutants. Perhaps UNC-8 exerts an indirect effect on dorsal assembly in this context in which remodeling is triggered prematurely when potential recycling of presynaptic components to nascent dorsal synapses could be limiting (*Park et al., 2011*).

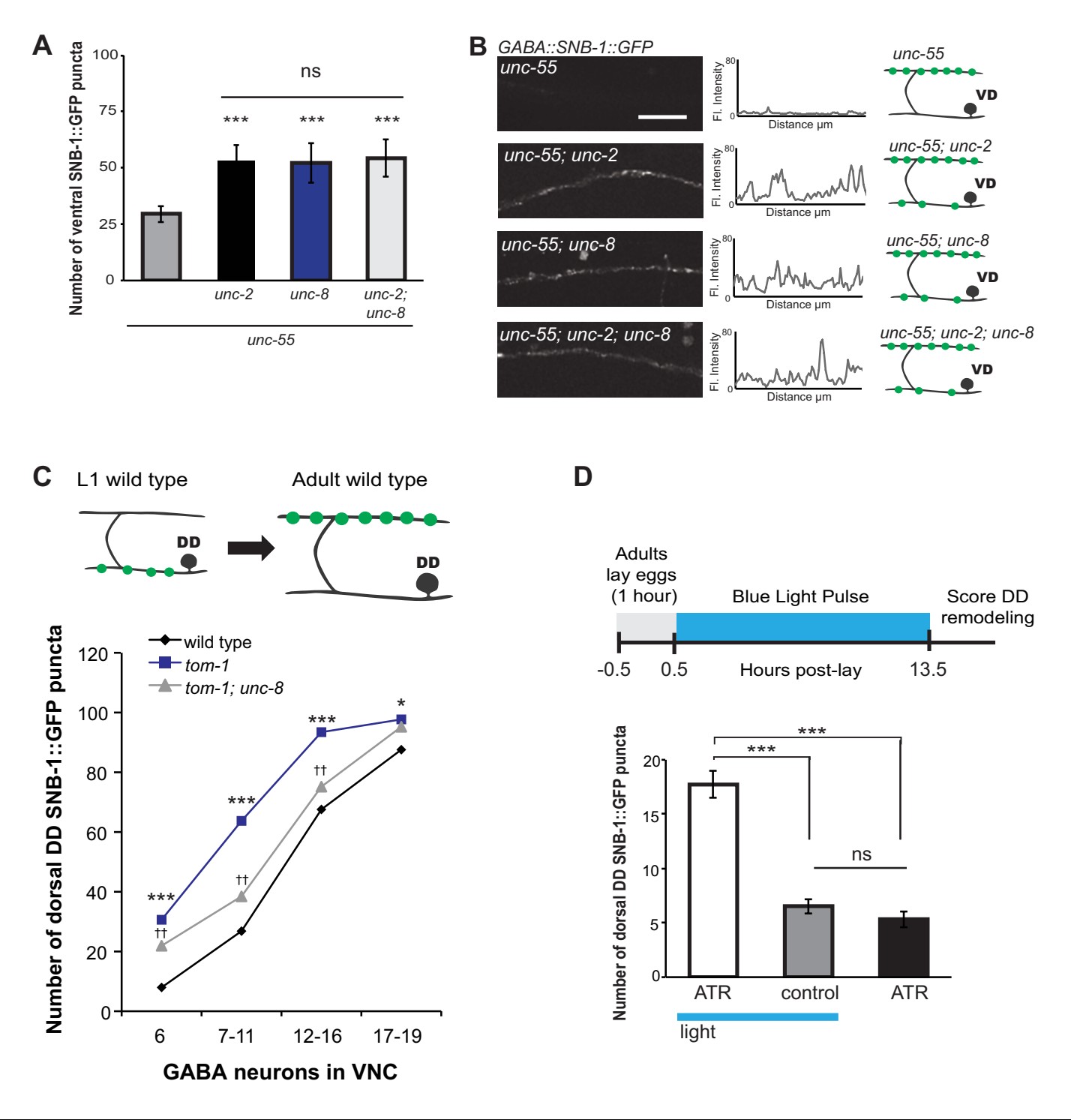

**Figure 6.** UNC-8 drives synaptic remodeling in an activity-dependent pathway that requires neurotransmitter release. (**A**) Loss-of-function mutations in either *unc-2* (VGCC) or *unc-8* impairs removal of ventral SNB-1::GFP in *unc-55* animals. The *unc-8* mutation does not enhance the *unc-55;unc-2* remodeling defect, demonstrating that UNC-2/VGCC and UNC-8/DEG/ENaC function in a common pathway to promote GABA synapse removal (***p<0.001 vs *unc-55*, One-Way ANOVA with Bonferroni correction, data are mean ± SD, n ≥ 17). (**B**) Representative images and insets show fluorescence intensity plots over a 20 μm region of the ventral nerve cord for each genotype. Scale bar is 5 μm. (**C**) DD synapses are precociously remodeled in *tom-1* mutants. This effect is suppressed in *unc-8; tom-1* animals (*p<0.05, ***p<0.001 *vs* wild type, ††p<0.01 vs *tom-1* Student's *t*-test, results pooled from ≥ 3 independent experiments per genotype) Plots are normalized for the total number of GABA neurons labeled with SNB-1::GFP in the ventral cord to account for developmental delay in *tom-1* mutants. (**D**) Optogenetic stimulation of channelrhodopsin (ChR2)-induced activity in

*Figure 6 continued on next page*

*Figure 6 continued*

GABA neurons for 13 hr (0.5 Hz) results in precocious appearance of SNB-1::GFP marked DD synapses in the dorsal nerve cord ($n \geq 18$ animals, ***p<0.001, data are mean ± SEM, One-Way ANOVA with Bonferroni correction), ATR is all-*trans* retinal. Mutant alleles were *unc-2(e55)* and *tom-1 (ok2437)*.

Our genetic results indicate that UNC-8 functions in an activity-dependent synaptic remodeling pathway. Because this conclusion is based on the analysis of mutants that alter neural activity throughout the *C. elegans* nervous system, we performed an experiment to ask if the GABA neuron activity alone is sufficient to drive synaptic remodeling. This requirement predicts that optogenetic activation of GABAergic neurons should accelerate the remodeling process. To test this idea, we used a GABA neuron-specific promoter to drive channelrhodopsin (ChR2) expression in DD neurons (*Liu et al., 2009*). Blue light exposure of these GABA::ChR2 transgenic animals evoked muscle relaxation with an immediate cessation of locomotion as expected for a treatment that enhances GABA release (data not shown) (*Schuske et al., 2004*). Synchronized populations of GABA::ChR2 animals were exposed to blue light at 0.5 Hz for 13 hr. DD neurons were assayed for precocious remodeling by counting the number of dorsal SNB-1::GFP puncta at the end of this period. The results of this experiment show that optogenetic activation of GABA neurons accelerates DD remodeling and that both blue light and exogenous all-*trans* retinal (ATR) are required for this effect; DD neurons that express ChR2 (GABA::ChR2) remodel earlier with significantly more dorsal puncta after 13 hr of light exposure in comparison to controls (*Figure 6D*). This finding argues that neuronal activity in DD neurons is sufficient to drive synaptic remodeling and thus suggests that a mechanism linking neural activity and *unc-8* function could be cell autonomous to GABA neurons.

## The calcium/calmodulin-dependent phosphatase calcineurin promotes synapse removal in the UNC-8 pathway

We have shown that calcium signaling through UNC-2 induces removal of ventral GABA synapses and that UNC-8 functions in this pathway to promote synapse disassembly (*Figure 6A*). In considering potential mechanisms for this effect, we performed a genetic experiment to test the idea that elevated intracellular calcium arising from UNC-2 function could activate cytoplasmic signaling proteins that drive synapse disassembly. One of these candidate downstream effectors, the calcium/calmodulin-activated phosphatase, calcineurin, is highly expressed throughout the nervous system and has been implicated in activity-dependent mechanisms that regulate synaptic maintenance and function (*Baumäartel and Mansuy, 2012*; *Winder et al., 1998*). Calcineurin is composed of two protein components, the catalytic subunit calcineurin A and the regulatory subunit calcineurin B. To determine if calcineurin function is required for the GABA neuron remodeling mechanism, we first tested the role of the *C. elegans* calcineurin A homolog, TAX-6 (*Figure 7A*). We found that a *tax-6* loss-of-function allele impedes the removal of ventral synapses in remodeling GABA neurons (*Figure 7B*). To ask if *tax-6* is expressed in GABA neurons, we determined that a translational reporter gene, *ptax-6::TAX-6::GFP* is co-localized with the GABA neuron-specific marker, *punc-47:: mCherry* (*Figure 7C*). We then used a loss-of-function allele of the calcineurin B homolog *cnb-1* to confirm the role of calcineurin in synaptic disassembly. Our results indicate that *cnb-1* function is required for the efficient removal of ventral synapses in remodeling GABA neurons. In addition, genetic ablation of *unc-8* in *unc-55; cnb-1* animals do not enhance this phenotype, which suggests that UNC-8 and calcineurin promote synapse elimination in a common pathway (*Figure 7D*).

These results indicate that calcineurin promotes the removal of ventral synapses and therefore predict that the constitutively active gain-of-function allele *tax-6(d)* should induce precocious synapse disassembly. To test this idea, we used the presynaptic marker GFP::RAB-3 to determine that DD ventral synapses are removed prematurely in *tax-6(d)* mutants; dorsal synaptic assembly is also precocious and occurs significantly earlier than in wild-type controls. Additionally, we determined that treatment of *tax-6(d)* animals with the DEG/ENaC inhibitor Benzamil significantly delays the precocious remodeling phenotype (*Figure 7E*). These results argue that intracellular calcium acting through TAX-6/Calcineurin promotes the overall synaptic remodeling mechanism and that this effect requires UNC-8 channel activity, thereby suggesting that *unc-8* functions downstream of *tax-6*.

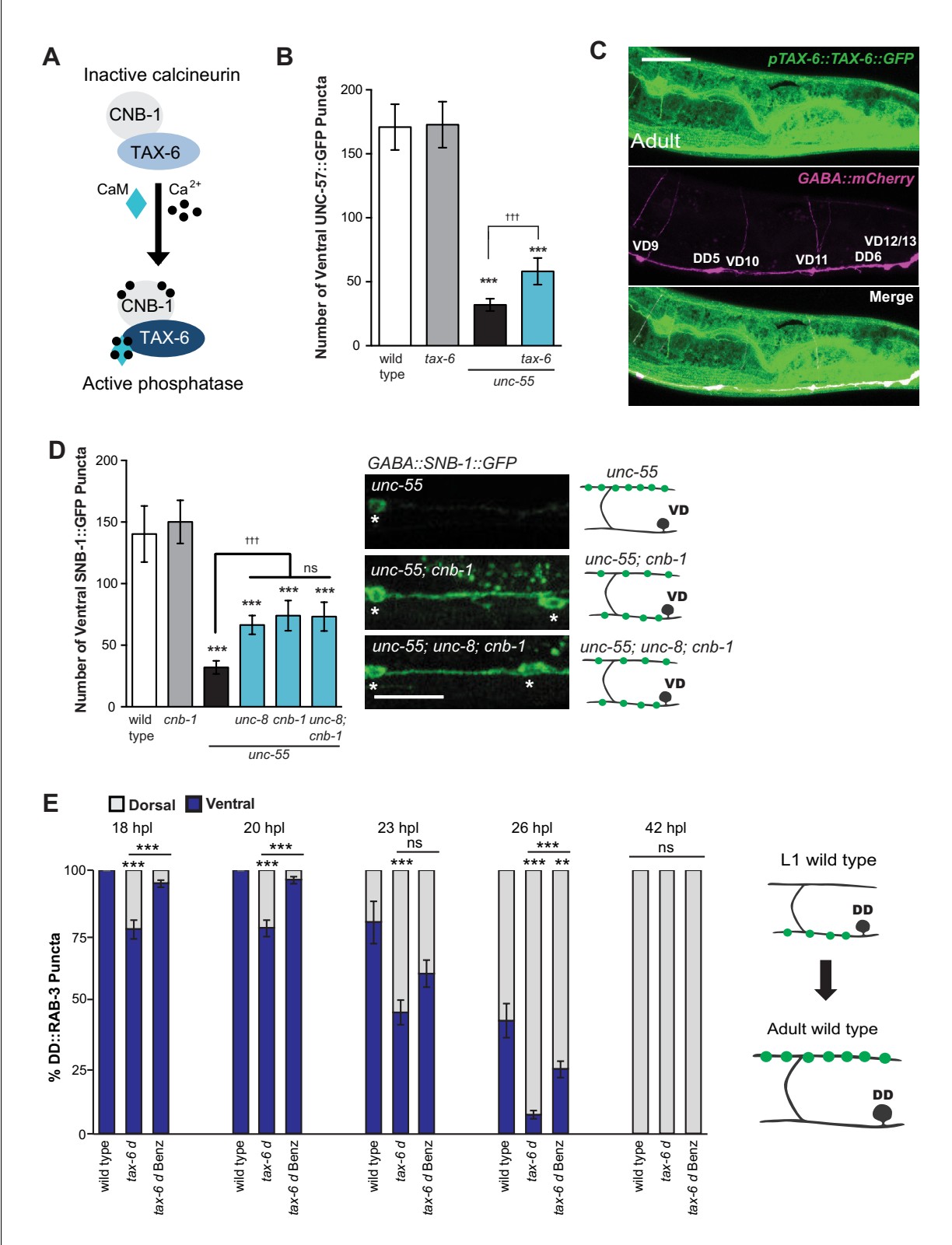

**Figure 7.** The calcium/calmodulin-dependent phosphatase calcineurin promotes GABA synapse removal in the UNC-8 pathway. (**A**) Calcineurin A and B subunits (TAX-6 and CNB-1, respectively) require calcium and calmodulin (CaM) to activate phosphatase activity. (**B**) Loss of *tax-6* partially suppresses the *unc-55* remodeling phenotype in GABA neurons (***p<0.001, ns is not significant, One-Way ANOVA with Bonferroni correction, data are mean ± SD, $n \geq 25$). (**C**) GFP-tagged TAX-6 under the control of the *tax-6* promoter region (*ptax-6::TAX-6::GFP*) is expressed in GABA neurons (*punc-47::*

*Figure 7 continued on next page*

*Figure 7 continued*

mCherry). Scale bar is 20 µm. (D) CNB-1 is required for remodeling in *unc-55* animals. Loss of *cnb-1* function does not enhance the *unc-55; unc-8* remodeling defect, suggesting that calcineurin and UNC-8 promote synapse removal in a common genetic pathway (left, ***p<0.001 compared to wild type, †††p<0.001 compared to *unc-55*, ns is not significant, One-Way ANOVA with Bonferroni correction, data are mean ± SD, wild type *n* = 10, mutants *n* = 20). Representative images of ventral nerve cords in *unc-55, unc-55; cnb-1* and *unc-55; unc-8; cnb-1* animals. Asterisks denote GABA neuron soma, scale bar is 20 µm (right). (E) Gain-of-function *tax-6 (tax-6d)* mutants remodel precociously and this effect is suppressed by Benzamil. Percentage of ventral (blue) vs dorsal (gray) DD synapses (*pflp-13::GFP::RAB-3*, **p<0.01, ***P<0.001, ns is not significant, One-Way ANOVA with Bonferroni correction, *n* ≥ 8 animals per timepoint, data are mean ± SEM). Results for *tax-6d* and for the *tax-6d* control for Benzamil treatment (see Materials and methods) are combined because they were not significantly different. Benz denotes 3mM DEG/ENaC inhibitor Benzamil. Mutant alleles were *tax-6 (p675), tax-6d(jh107), cnb-1(ok276)*.

To summarize, our findings are consistent with the hypothesis that neuronal activity drives synaptic remodeling through a calcium-dependent signaling pathway in which calcineurin and *unc-8* function together to promote disassembly of the presynaptic apparatus.

## UNC-8 promotes synapse elimination in a common pathway with the cell death gene, CED-4

Previous work has shown that the canonical apoptotic pathway promotes the removal of ventral GABA synapses in remodeling DD neurons. In this mechanism, the adaptor protein CED-4/Apaf1 and its downstream effector, CED-3/caspase activate the actin-severing protein gelsolin to destabilize a presynaptic F-actin network during synapse elimination (*Meng et al., 2015*). Because synaptic removal is activated during a discrete developmental period, a specific signal is likely required to trigger this pathway. Based on our finding that calcium signaling and UNC-8 function together to promote GABA synapse elimination and previous work that detected roles for calcium in caspase activation and gelsolin function (*Pinan-Lucarre et al., 2012*; *Liu et al., 2011*), we considered the possibility that these components function in a common pathway and that calcium activates the overall mechanism. To test this idea, we first asked if the cell death gene *ced-4* is required for GABA synapse removal in *unc-55* mutants in which both DD and VD neurons remodel. A loss-of-function mutation in *ced-4* results in significant retention of ventral GABA synapses in comparison to *unc-55* mutant animals and therefore confirms that *ced-4* is necessary for the efficient removal of ventral synapses in both classes (*e.g.*, DD and VD) of remodeling GABA neurons (*Figure 8A*, [*Meng et al., 2015*]). The *ced-4*-dependent effect on remodeling in *unc-55* animals is comparable to that of *unc-55; unc-8* double mutants. Furthermore, the combination of *ced-4* and *unc-8* mutants did not further enhance the restoration of ventral synapses over that of either *unc-55; unc-8* or *unc-55; ced-4* animals (*Figure 8A*). Together, these results suggest that the DEG/ENaC protein UNC-8 functions in a common pathway with *ced-4* to eliminate ventral GABA synapses. Because we have also shown that calcium and *unc-8* function in a common pathway to promote removal of GABA synapses, we propose that a rise in intracellular calcium in active GABA neurons is sufficient to trigger *ced-4*-dependent synapse elimination (*Figure 8B*, see Discussion).

## Discussion

Developing nervous systems are dynamically remodeled during genetically defined intervals in which circuit architecture is particularly sensitive to neural activity. These 'critical periods' have been widely observed, but the underlying mechanisms that account for the dual roles of transcriptional pathways and synaptic activity in circuit refinement are incompletely defined. In this study, we show that a member of the conserved DEG/ENaC family of cation channel proteins, UNC-8, is transcriptionally controlled to mediate an activity-dependent pathway that promotes the removal of GABAergic synapses in *C. elegans*.

## The UNC-8/DEG/ENaC protein promotes synapse elimination in an activity-dependent pathway

Previous work has detected roles for DEG/ENaC proteins in synaptic function and neural plasticity. For example, members of the ASIC (acid-sensing ion channels) class of the DEG/ENaC family

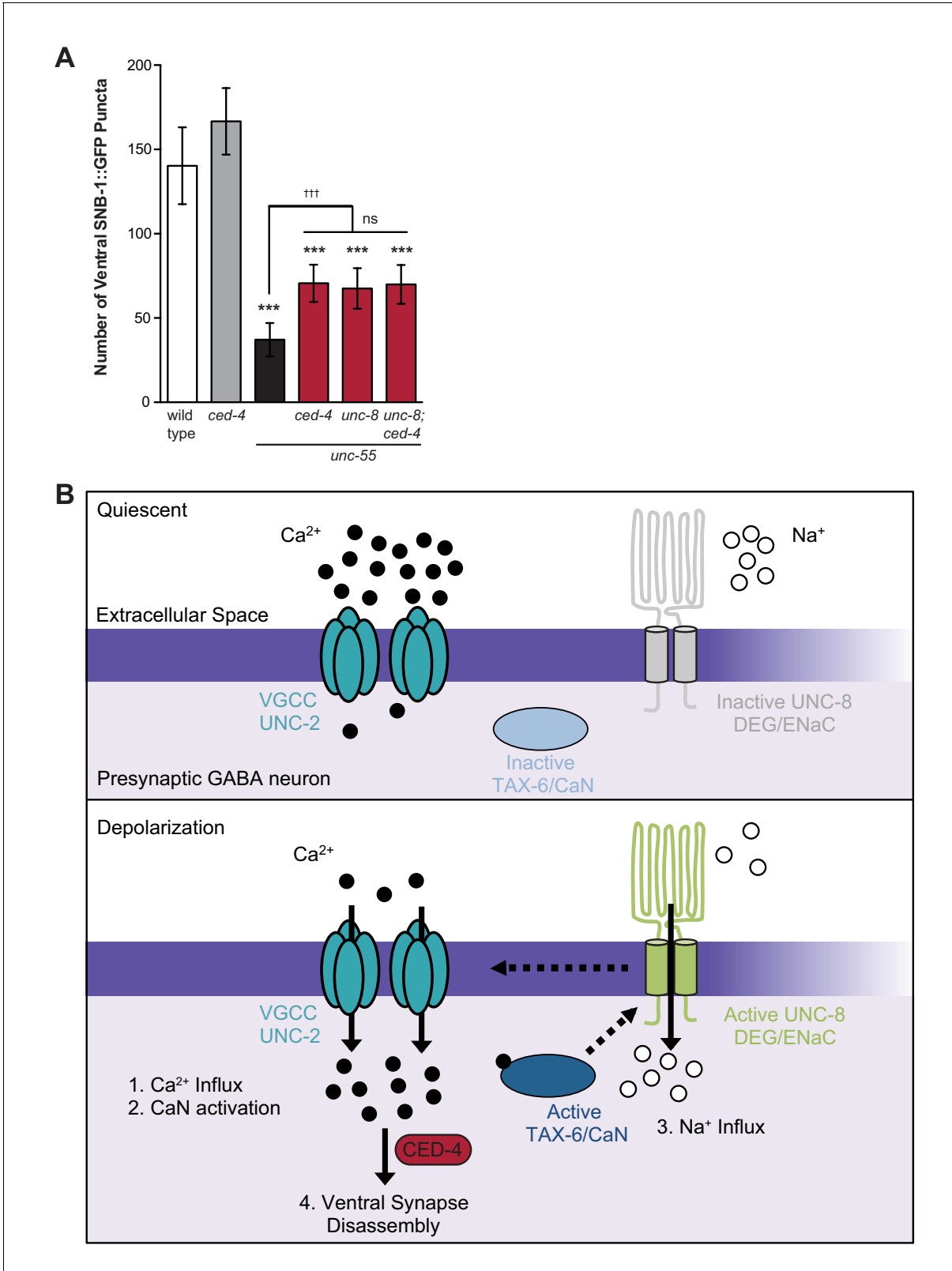

**Figure 8.** Model of UNC-8-driven synapse disassembly in the GABA neuron remodeling program. (**A**) A loss-of-function mutation in the pro-apoptotic gene *ced-4* partially suppresses GABA neuron remodeling in *unc-55* animals. Genetic ablation of *unc-8* in the triple mutant *unc-55; ced-4; unc-8* mutants does not enhance the remodeling defect and thus suggests that UNC-8 and CED-4 promote synapse elimination in a common pathway (***p<0.001 compared to wild type, †††p<0.001 compared to *unc-55*, ns is not significant, One-Way ANOVA with Bonferroni correction, data are mean ±

*Figure 8 continued on next page*

*Figure 8 continued*

SD, wild type *n* = 10, mutants *n* $\geq$ 20) Mutant allele is *ced-4(n1162)*. (**B**) Model of a calcium-dependent mechanism for removal of the presynaptic apparatus. A DEG/ENaC channel containing UNC-8 is not active (gray) in quiescent GABA motor neurons (top panel). (1) GABA neuron depolarization activates the voltage-gated calcium channel (VGCC), UNC-2, to allow calcium entry (black circles bottom panel). (2) Intracellular calcium activates the calcium/calmodulin-dependent phosphatase, calcineurin (CaN/TAX-6). (3) CaN phosphatase may activate UNC-8, which results in the movement of sodium ions (white circles) into the presynaptic GABA neuron, further depolarizing the presynaptic membrane and activating VGCCs. This positive feedback loop is predicted to further elevate intracellular calcium. (4) Our results show that UNC-8 drives the removal of presynaptic components and functions in a common genetic pathway with calcium signaling and with the apoptotic protein CED-4. Therefore, we hypothesize that selective expression of UNC-8 in remodeling GABA neurons effectively boosts the level of intracellular calcium to activate a CED-4-dependent pathway for removal of the presynaptic apparatus.

proteins are activated by low pH and have been implicated in learning and memory (*Wemmie et al., 2002*, *2004*; *Zha et al., 2006*). In one possible mechanism to explain this role, acidification of the synaptic cleft that accompanies the neurotransmitter release is thought to trigger ASIC-mediated sodium influx. The resultant membrane depolarization is proposed to enhance postsynaptic signaling by activating local voltage gated ion channels and NMDA receptors (*Wemmie et al., 2013*). Presynaptic roles are also possible as suggested by studies in *C. elegans* that detected an ASIC-like channel that promotes neurotransmitter secretion in a learning and memory paradigm (*Voglis and Tavernarakis, 2008*). Our finding that a reconstituted UNC-8 channel is in fact inhibited by low pH, however, suggests that UNC-8 is unlikely to function as an ASIC-like protein in vivo (*Wang et al., 2013*). This observation is important because non-ASIC members of the DEG/ENaC family are also expressed in the brain, but their roles in synaptic function are largely unknown (*Giraldez et al., 2013*; *Yamamura et al., 2004*; *Waldmann, 1995*). A recent study established that the DEG/ENaC protein, pickpocket, exerts a homeostatic function that maintains neurotransmitter release at the *Drosophila* neuromuscular junction. In this mechanism, sodium influx arising from elevated pickpocket transcription and membrane insertion is proposed to result in local depolarization that activates a nearby voltage-gated calcium channel (VGCC) and consequent neurotransmitter release (*Younger et al., 2013*). This model is consistent with the finding that the $Ca_V2.1$ VGCC channel subunit cacophony is required for synaptic homeostasis (*Müller and Davis, 2012*). Our results are suggestive of a similar presynaptic role for UNC-8 of enhancing intracellular calcium influx with the important difference that UNC-8 ultimately eliminates neurotransmitter secretion by triggering destruction of the presynaptic apparatus. In addition, our findings provide a plausible explanation for the radically different outcomes of pickpocket and UNC-8 channel function.

In this study, we demonstrate that UNC-8 promotes the removal of presynaptic domains in remodeling GABAergic neurons. This UNC-8 function requires its intrinsic cation channel activity (*Figure 5*) which likely involves sodium influx since electrophysiological studies of a reconstituted UNC-8 channel in *Xenopus* oocytes indicate that UNC-8 preferentially transports sodium and is impermeant to calcium (*Wang et al., 2013*). Because our evidence points to punctate UNC-8 localization within the ventral nerve cord (*Figure 2F*), it is reasonable to predict that UNC-8-dependent sodium import would activate nearby VGCC channels (*Wemmie et al., 2013*). This idea is consistent with the observation that VGCC function is required for ASIC-induced elevation of intracellular calcium in hippocampal neurons (*Zha et al., 2006*). Furthermore, our results show that synaptic removal in remodeling *C. elegans* GABA neurons is disrupted by genetic ablation of the $Ca_V2$ channel subunit, UNC-2. As a key component of the presynaptic apparatus, UNC-2 mediates calcium influx to drive neurotransmitter release (*Richmond et al., 2001*). We suggest that the UNC-2-dependent rise in cytoplasmic calcium in active GABA neurons is enhanced by UNC-8 cation channel function. Moreover, because our genetic results indicate that *unc-2* and *unc-8* function in a common pathway (*Figure 6A*), the elevation of intracellular calcium arising from this interaction could be required for efficient removal of the presynaptic apparatus in remodeling GABA neurons (see below).

## Calcium signaling and a caspase-dependent mechanism drive GABA neuron synapse elimination

The idea of a downstream role for calcium in synaptic remodeling is underscored by our finding that the calcineurin homolog, TAX-6, also functions in the *unc-8*-dependent mechanism of synapse

elimination (*Figure 7*). The serine-threonine phosphatase activity of calcineurin/CaN is calcium and calmodulin-dependent and has been shown to regulate neural plasticity. For example, calcineurin/CaN antagonizes LTP in a mechanism that dephosphorylates the GluR1 subunit of the AMPA receptor to promote its removal from the postsynaptic membrane (*Winder et al., 1998*; *Gorski et al., 2012*; *Lee et al., 1998*). TAX-6/Calcineurin could exert a parallel role in the removal of the presynaptic apparatus in GABA neurons by activating the UNC-8 channel. Our finding that the DEG/ENaC inhibitor, Benzamil, potently blocks the accelerated remodeling phenotype of a constitutively active TAX-6 protein is consistent with this idea (*Figure 7E*). The resultant positive feedback loop involving UNC-2, TAX-6/Calcineurin and UNC-8 should further elevate intracellular calcium (*Figure 8*). In an analogous mechanism, phosphorylation by CaMKII activates ASIC1a and the resultant VGCC-dependent elevation of intracellular calcium in turn promotes CaMKII activity (*Zha et al., 2006*; *Gao et al., 2005*; *Zha and Zha, 2013*). In this model, UNC-8 would act as a synaptic amplifier to drive a self-reinforcing signaling pathway that elevates intracellular calcium.

The key role of calcium signaling in this model is consistent with previous work showing that removal of the GABA presynaptic apparatus requires components of the canonical apoptotic pathway including the caspase-3 homolog CED-3 and its upstream activator CED-4/Apaf1. In this mechanism, the CED-3 protease cleaves the actin-severing protein gelsolin to release an active gelsolin domain that in turn destabilizes an F-actin network at the presynaptic membrane (*Meng et al., 2015*). Notably, both CED-3 and gelsolin function in calcium-dependent pathways (*Pinan-Lucarre et al., 2012*; *Liu et al., 2011*). Because our genetic evidence argues that *unc-8* and *ced-4* function in a common pathway (*Figure 8A*), we propose a model in which UNC-8 promotes the elevation of intracellular calcium above a critical threshold that then triggers the apoptotic pathway to dismantle the presynaptic region. The localization of apoptotic components to the presynaptic membrane (*Meng et al., 2015*) may ensure that a local rise in calcium near the synapse is sufficient to activate this mechanism, but also protects the neuron from apoptotic death. The conserved role of CED-3/caspase-3 in synapse elimination in other species (*Ertürk et al., 2014*; *Wang et al., 2014*; *Li et al., 2010*) argues that similar DEG/ENaC-dependent mechanisms could be widely employed to trigger synaptic destruction.

## Multiple components regulate GABA neuron synaptic remodeling

Our results show that *unc-8* promotes the removal of ventral synapses in remodeling DD and VD motor neurons, but is generally not necessary for the nascent assembly of dorsal synapses (*Figure 2E*). The proposed downstream role of the apoptotic pathway is similarly limited to synaptic removal (*Meng et al., 2015*). In addition, the elimination of GABAergic synapses is only partially dependent on this *unc-8* pathway (*Figure 2C*, *Figure 3*, *Figure 3—figure supplement 1A*, *Figure 4*) (*Meng et al., 2015*). These findings point to important roles for additional components in GABA neuron synaptic remodeling (*Howell et al., 2015*; *He et al., 2015*). For example, the cyclin box-containing protein CYY-1 and cyclin-dependent kinase CDK-5 function together to promote synaptic removal and reassembly, respectively (*Park et al., 2011*). In previous work, we have shown that the Iroquois family homeodomain transcription factor, IRX-1, promotes synaptic removal. IRX-1 also promotes dorsal DD synapse formation, suggesting that IRX-1 orchestrates expression of multiple genes that drive DD synaptic remodeling (*Petersen et al., 2011*). The exact function of the IRX-1 pathway in synaptic disassembly is unknown, but likely involves the extraction of components that are required for GABA release since functional GABA synapses are restored to the ventral nerve cord of *unc-55* animals that are also mutant for *irx-1* (*Petersen et al., 2011*). In the future, it will be interesting to define the specific roles of each of these effectors in the mechanism of GABA neuron synaptic remodeling. Despite the necessity for additional pathways functioning in parallel to *unc-8* to execute a comprehensive GABA neuron remodeling program, our results also determined that UNC-8 alone is sufficient to trigger synapse elimination (*Figure 3—figure supplement 2B*). This finding argues that transcriptional regulation of *unc-8* expression effectively functions as a genetic switch that determines the developmental fate of the presynaptic signaling apparatus in active GABAergic neurons.

# Materials and methods

### *C. elegans* strains and genetics

Strains were maintained at 20°C on NGM plates seeded with OP50 (*E. coli*), unless otherwise stated (*Brenner, 1974*). The wild type strain is N2 and only hermaphrodite animals were analyzed. The *unc-55(e1170)* and *unc-8(tm5052)* alleles were used for these studies. For a complete list of the strains used in this study, see *Supplementary file 1A*.

The *unc-8(tm5052)* allele was obtained from a UV/TMP mutagenized library, as described previously (*Gengyo-Ando and Mitani, 2000*) and was identified by PCR amplification with primers spanning the deleted region. *tm5052* likely corresponds to an *unc-8* null allele because it deletes a portion of the fifth and entire sixth exon (197 base pairs) with the insertion of CT resulting in a premature stop codon prior to the first transmembrane domain. We used the following primers to detect the *tm5052* mutation: Forward 5'-TGGGGCCCTAATAATTTCGA-'3 and Reverse 5'- AGTGA-CAGTATGAAGCCAGG-'3.

### Molecular biology and transgenes

Construction of GABA csRNAi transgenic lines

RNAi plasmids used for GABA neuron-specific knock down of *unc*-8 target the first 7 exons of the *unc-8* coding region. To clone the *unc-8* sense construct pSA76, a 2.3 kb region of *unc-8* cDNA was amplified with the following primers containing 5'Ascl/3'SacII adaptors: Forward 5'- GGCGCGCCA TGTCACCTTTGCTGACGT-3' and Reverse 5'-GCCAGGAGGTGATATTCTAGCCGCGG-3'. This fragment was cloned into pCR2.1 via TOPO-TA reaction (Invitrogen, Waltham, MA) to yield pSA75. The 2.3 kb *unc-8* cDNA fragment was then subcloned into the existing GABAergic cell-specific RNAi (csRNAi) plasmid pSA47 via Ascl/SacII to yield pSA76 (*Petersen et al., 2011*). pSA76 contains the DD/VD specific promoter, *pttr-39* and the *unc-119* wild-type mini gene (*Maduro and Pilgrim, 1995*). To construct the *unc-8* antisense plasmid pSA78, the 2.3 kb *unc-8* cDNA fragment was amplified with the following primers containing 5''SacII/3'Ascl adaptors: Forward 5'-CCGCGGATGTCACC TTTGCTGACGTG-3' and Reverse 5'- CCAGGAGGTGATATTCTAGGGCGCGCC-3'. The *unc-8* antisense fragment was subcloned into pCR2.1 via TOPO-TA reaction (Invitrogen) to yield pSA73. The *unc-8* fragment from pSA73 was then inserted into the GABA neuron-specific RNAi (csRNAi) plasmid pSA47 via Ascl/SacII to yield pSA77. The *unc-8*-containing region of pSA77 between Scal and SacII was then inserted into plasmid pSA49 to yield pSA78. The *pttr-39* promoter in pSA78 drives expression of the 2.3 kb *unc-8* antisense fragment and mCherry. pSA76 (*unc-8* sense) and pSA78 (*unc-8* antisense) were linearized and ligated, then transformed into *unc-119* worms via microparticle bombardment to yield a spontaneous integrant (strain NC2601) as indicated by 100% transmission of rescued (*unc-119+*) movement (indicating *unc-8* sense) and mCherry expression in all GABAergic motor neurons (indicating *unc-8* antisense) (*Praitis et al., 2001*). A control plasmid was also created containing pttr39-driven mCherry and the *unc-119* rescuing gene, which was transformed into *unc-119* worms via microparticle bombardment.

### Recombineering UNC-8::GFP fosmid

The UNC-8::GFP fosmid was recombineered as previously described (*Tursun et al., 2009*). Briefly, the 30 kb fosmid WRM0635cA02 containing the *unc-8* genetic locus was obtained from GeneService (Nottingham, UK) and purified. Fosmid DNA was transformed into electrocompetent SW105 cells and was verified by PCR. A GFP-galK recombineering cassette was amplified with 50 kb homology arms from pBALU1 and gel-purified. The GFP-galK PCR product was transformed into electrocompetent, λRed recombinase-activated, fosmid-containing SW105 cells. The cells containing the fosmid and GFP-galK were grown for more than 60 hr at 32°C and streaked on MacConkey and galactose plates with chloramphenicol to ensure the insertion of recombineering cassette. To excise galK from the GFP intron, colonies were incubated with 0.1% arabinose to create an *unc-8::GFP* expression fosmid. This *unc-8::GFP* fosmid was then purified and confirmed by sequencing. The fosmid was injected into *unc-8(tm5052)* animals at 25 ng/µl with co-injection marker *pceh-22::*GFP at 15 ng/µl.

## Construction of *punc-25*::UNC-8::GFP plasmid

UNC-8 cDNA was PCR-amplified from pSGEM/pTWM60 (*Wang et al., 2013*) with primers that span the UNC-8 cDNA sequence and exclude the 3' stop codon. The primer sequences are: Forward 5'-A TGAGCGCAAGGAGTAGT-3' and Reverse 5'- TTTGCTCATTAACTCCTTTGT-3'. Primers include either 5'-AscI or 3'-SacII adaptors for inserting UNC-8 cDNA into pMLH260 (*punc-25::coq1cDNA:: GFP::unc-54*) in place of the *coq-1* fragment. The resultant plasmid, pTWM62 (*punc-25*::UNC-8:: GFP) was injected (10 ng/µl) with co-selectable marker *pmyo-2*::mCherry::*unc-54* (2.5 ng/µl) into *unc-8 (tm5052)* animals.

## Construction of *pttr-39*::UNC-8 plasmid

UNC-8 cDNA was PCR-amplified from pSGEM/pTWM60 (*Wang et al., 2013*) with primers that span the UNC-8 cDNA sequence. The primer sequences are: Forward 5'-ATGAGCGCAAGGAGTAGT-3' and Reverse 5'- TTTGCTCATTAACTCCTTTGT-3'. Primers include either 5'-AscI or 3'-EcoRI adaptors for inserting UNC-8 cDNA into pTWM35 (*pttr39::arx-5::GFP::unc-54*) in place of the ARX-5::GFP fragment. The resultant plasmid, pTWM92 (*pttr-39*::UNC-8cDNA) was injected (25 ng/µl) with co-selectable markers *pmyo-2*::mCherry::*unc-54* (2 ng/µl) and *punc-25*::mCherry::RAB-3 (5 ng/µl) into *unc-55; unc-8 (tm5052) juIs1* or *juIs1* animals.

## Staging and synapse quantification

For time-course experiments, 100 adult hermaphrodites from each genotype were picked to fresh 60 mm plates and allowed to lay eggs for one hour. The mid-point at which the eggs were laid is considered $T_0$. All adults were removed from the plates after 1 hr. Plates were maintained at 23°C until assayed. Puncta arising from localization of fluorescent presynaptic markers were counted with a Zeiss Axiovert microscope (63X oil objective) in immobilized animals. For timecourse experiments puncta were counted between DD1 and DD6 (*Figures 2C*, *6C,D*, *7E*, *Figure 2—figure supplement 1B–D*) or from VD3 to VD11 in adults (*Figure 3—figure supplement 1B*). Data were pooled from 3 separate experiments. In young adults, labeled puncta were counted in the ventral nerve cord region between VD3 and VD11 (*Figures 2E*, *4A–E*, *5B*, *6A*, *7B,D*, *8A*, *Figure 1—figure supplement 1C*, *Figure 3—figure supplement 1*, *Figure 5—figure supplement 1A*). For experiments featuring mosaic expression of either *unc-8(csRNAi)* (*Figure 3F*) or *unc-8cDNA* (*Figure 3—figure supplement 2*), puncta were counted from individual DD and VD neurons. The examiner was blinded to genotype.

## Confocal microscopy and image analysis

Animals were immobilized with 15 mM levamisole/0.05% tricaine and mounted on a 2% agarose pad in M9 buffer as previously described (*Smith et al., 2010*). Z-stack images were collected on a Leica TCS SP5 confocal microscope using a 63X oil objective (0.5 µm/step), spanning the focal depth of the ventral nerve cord GABA neurons and synapses. Leica Application Suite Advanced Fluorescence (LAS-AF) software was used to generate maximum intensity projections. Images in *Figure 3A* were collected from 10 animals of each genotype. Ventral nerve cord images between VD4 and VD5 were straightened using an ImageJ plug-in and aligned in rows. All fluorescence intensity plots were created by drawing a line through the ventral nerve cords of each animal and calculating the fluorescence intensity value in arbitrary units over the distance in micrometers with the ImageJ plot profile tool. For *Figure 1*, DD and VD neurons were identified by the GABA-specific marker *pttr-39:: mCherry*. The cells were traced in ImageJ and the background was subtracted. The *unc-8::GFP* fluorescence intensity was then normalized to the *pttr-39::mCherry* fluorescence intensity for each cell. Fluorescence intensity plots in *Figure 4* and *Figure 4—figure supplement 1* were created with the ImageJ plot profile tool, analyzing the same region of the ventral nerve cord in both GFP and RFP channels. Fluorescence intensity values were normalized for each channel. The coefficient of determination ($r^2$) was calculated in ImageJ using the Manders coefficients macro, from at least 10 animals for each genotype. $r^2$ values were averaged and presented as mean ± SEM. An $r^2$ value of 0 represents no co-localization, whereas $r^2 = 1$ represents complete co-localization.

## Pharmacology

Amiloride hydrochloride hydrate (Sigma, #A7410) stock solution was prepared in sterile water (50 mg/ml) and stored at -20°C. A final concentration of 3 mM Amiloride diluted in OP50 bacteria was seeded on NGM plates. Control NGM plates contained the same volume of sterile water added to OP50. Benzamil hydrochloride hydrate (Sigma, #B2417, St. Louis, MO) stock solution was prepared in sterile water (1 mg/ml) and stored at 4°C. A final concentration of 3 mM Benzamil diluted in OP50 bacteria was seeded on NGM plates. Control NGM plates contained the same volume of sterile water added to OP50. Plates were stored at 4°C for up to one week. Five adult *unc-55; juIs1* animals were placed on either Amiloride, Benzamil, or control plates at room temperature and progeny examined at the young adult stage (*Figure 5A,B* and *Figure 5—figure supplement 1A*). Adult *tax-6 (d); wyIs202* animals were grown on Benzamil or control plates and their larval progeny were collected on control or Benzamil plates for timecourse assays (*Figure 7E*). The number of ventral puncta was counted using a Zeiss Axiovert microscope (63X oil objective) and Z-stack images were captured on a Leica TCS SP5 confocal microscope using a 63X oil objective (0.5 µm/step). The examiner was blinded to genotype and treatment condition.

## Optogenetics

All-*trans* retinal (Sigma, #R2500) was dissolved in ethanol to prepare a 100 mM stock and stored at -20°C. 300 µM of all-*trans* retinal stock solution (ATR plates) or ethanol (control plates) was added to OP50 bacteria and seeded onto NGM plates. Plates were protected from light and were stored at 4°C for up to one week. 100 adult hermaphrodites were placed on either ATR or control plates, allowed to lay eggs for 1 hr and then removed from the plate. The midpoint of this hour is considered $T_0$. Plates were exposed to blue light pulses with a 470-nm LED light (#M470L2, Thor Labs, Newton, NJ) for 13 hr (0.5 Hz, 2 mW/mm$^2$ measured with Solartech Inc. Solar Meter 9.4 radiometer). Light stimulation was controlled using NI Max software through TTL signals generated by a digital function generator (National Instruments, Austin, TX). 13 hr after egg laying, animals were assayed for DD remodeling by counting the number of dorsal SNB-1::GFP puncta. The examiner was blinded to the treatment. Data were collected from three independent time course experiments with at least six animals per treatment.

## Electrophysiology

The *C. elegans* dissection and electrophysiological methods were as previously described (*Richmond and Jorgensen, 1999*). Animals were immobilized along the dorsal axis with Histoacryl Blue glue, and a lateral cuticle incision was made with a hand-held glass needle, exposing ventral medial body wall muscles. Muscle recordings were obtained in the whole-cell voltage-clamp mode using an EPC-10 patch-clamp amplifier and digitized at 1 kHz. The extracellular solution consisted of 150 mM NaCl, 5 mM KCl, 5 mM CaCl$_2$, 4 mM MgCl$_2$, 10 mM glucose, 5 mM sucrose, and 15 mM HEPES (pH 7.3, ~340 mOsm). The low Cl intracellular patch pipette solution used to isolate outward GABA minis at a 0 mV holding potential was composed of 115 mM KGluconate, 25 mM KCl, 0.1 mM CaCl$_2$, 1 mM BAPTA and 50 mM HEPES. Data were acquired using Pulse software (HEKA, Southboro, Massachusetts, United States) run on a Dell computer. Subsequent analysis and graphing was performed using Pulsefit (HEKA), Mini analysis (Synaptosoft Inc., Decatur, Georgia, United States) and Igor Pro (Wavemetrics, Lake Oswego, Oregon, United States).

## Oocyte expression and electrophysiology

UNC-8(G387E) cRNA was synthesized using T7 mMESSAGE mMACHINE kit (Ambion, Waltham, MA). cRNA was purified and examined on a denaturing agarose gel to confirm correct size and integrity. cRNA quantification was performed spectroscopically. Stage VI defolliculated oocytes from *Xenopus Laevis* were purchased from Ecocyte Bioscience US LLC (Austin, Texas). Oocytes were injected with 10 ng/oocyte of cRNA and incubated in OR2 solution (82.5 mM NaCl, 2.5 mM KCl, 1 mM CaCl$_2$, 1 mM MgCl$_2$, 1 mM Na$_2$HPO$_4$, 0.5 g/liter polyvinyl pyrolidone, and 5 mM HEPES, pH 7.2, supplemented with penicillin and streptomycin (0.1 mg/ml) and 2 mM Na-pyruvate) plus 500 µM amiloride (to prevent channel hyperactivation-dependent cell death) at 20°C for 2–3 d before recordings. Currents were measured using a two-electrode voltage-clamp amplifier (Gene-Clamp 500B; Axon Instruments, Sunnyvale, CA) at room temperature. Electrodes (0.2–0.5 MΩ) were

filled with 3 M KCl, and oocytes were perfused with a physiological NaCl solution (100 mM NaCl, 2 mM KCl, 1 mM CaCl$_2$, 2 mM MgCl$_2$, and 10 mM HEPES, pH 7.2) and divalent cation free plus EGTA NaCl solution (110 mM NaCl, 2 mM KCl, 1 mM EGTA, and 10 mM HEPES, pH 7.2). pH was adjusted at the indicated values using NaOH. The oocyte membrane was clamped at −30 mV and stepped from −160 to +100 mV. Benzamil was added to the solutions from a stock of 10 mM. A saturating concentration of benzamil (1mM) was added at the end of each experiment to confirm that endogenous/leak currents were similar in amplitude to non-injected oocytes within each oocyte batch. Oocytes that had larger endogenous/leak currents were not further analyzed. We used the pCLAMP suite of programs (Axon Instruments) for data acquisition and analysis. Currents were filtered at 200 Hz and sampled at 1 kHz. We used OriginPro 8 (OriginLab Corporation, Northampton, MA) to generate graphs, K$_i$, and for statistical analysis.

## Electron microscopy

Young adult hermaphrodites of each strain were prepared for high-pressure freeze (HPF) fixation as described (*Rostaing et al., 2004*). 10–15 animals were loaded into a specimen chamber filled with *E. coli*. The specimens were frozen rapidly in a high-pressure freezer (Bal-Tec HPM010) at −180°C and high pressure. Freeze substitution was performed on frozen samples in a Reichert AFS machine (Leica, Oberkochen, Germany) with 0.1% tannic acid and 2% OsO$_4$ in anhydrous acetone. The temperature was kept at −90°C for 107 hr, increased at 5°C/hr to −20°C, and kept at -20°C for 14 hr. The temperature was then increased by 10°C/h to 20°C. Fixed specimens were embedded in Epon resin after infiltration in 50% Epon/acetone for 4 hr, 90% Epon/acetone for 18 hr, and 100% Epon for 5 hr. Embedded samples were incubated for 48 hr at 65°C. All specimens were prepared in the same fixation procedure and labeled with anonymous tags so that the examiner was blinded for genotype. Ultra thin (40 nm) serial sections were cut using an Ultracut 6 (Leica) and collected on formvar- covered, carbon-coated copper grids (EMS, FCF2010-Cu). Grids were counterstained in 2% aqueous uranyl acetate for 4 min, followed by Reynolds lead citrate for 2 min. Images were obtained on a Jeol JEM-1220 (Tokyo, Japan) transmission electron microscope operating at 80 kV. Micrographs were collected using a Gatan digital camera (Pleasanton, CA) at a magnification of 100k. Images were quantified using NIH ImageJ software. Dorsal and ventral cords were distinguished by size and morphology. GABAergic synapses were identified by previously established criteria, including position in the cord as well as the morphology of the synapse (*White et al., 1986*; *Jin et al., 1999*). GABAergic synapses are larger than their cholinergic motor neuron counterparts, and the active zones in these synapses form a direct, perpendicular angle with muscle arms. On the other hand, the presynaptic density in cholinergic synapses orient at an acute angle to the muscle, generally 30–45° and are often dyadic. Some images were collected at 30 k to aid in identifying synaptic identity based on terminal position in the cord. Two colleagues with expertise in EM reconstruction of the *C. elegans* ventral nerve cord independently reviewed synapse images from each strain to verify identification. Each profile represents an image taken of a 40 nm section. A synapse was defined as a set of serial sections containing a presynaptic density and flanking sections from both sides without presynaptic densities. Synaptic vesicles were identified as spherical, light gray structures with an average diameter of ~30 nm. At least two animals were analyzed for each genotype. Numbers of profiles analyzed for each genotype were: wild type = 502, *unc-8* = 322, *unc-55* = 246, *unc-55; unc-8* = 304 for ventral GABAergic synapse evaluation; wild type = 502, *unc-8* = 322, *unc-55* = 246, *unc-55; unc-8* = 304 for ventral cholinergic synapse evaluation.

## Acknowledgements

We thank MQ Dong for MQD5, M Francis for IZ1607, O Hobert for *otEx2876*, Y Jin for *juIs137*, E Jorgensen for *oxIs351*, J Kaplan for *nuIs279*, K Shen for *wyIs202*, M Zhen for *hpIs3*, M Zhen and M Kittelman for advice on scoring GABA synapses by electron microscopy and members of the Miller Lab for discussions. This work made use of instruments in the Electron Microscopy Service (Research Resources Center, UIC). Some strains used in this study were provided by the CGC, which is funded by the NIH Office of Research Infrastructure Programs (P40 OD010440). We also received strains from the Japanese National BioResource Project. This work was supported by NIH grants 5R01NS081259 (DMM), 1F31NS084732 (TWM), 1F31NS063669 (SCP).

## Additional information

### Funding

| Funder | Grant reference number | Author |
|---|---|---|
| National Institute of Neurological Disorders and Stroke | 1F31NS084732 | Tyne W Miller-Fleming |
| National Institute of Neurological Disorders and Stroke | 1F31NS063669 | Sarah C Petersen |
| National Institute of Neurological Disorders and Stroke | 5R01NS081259 | David M Miller III |

The funders had no role in study design, data collection and interpretation, or the decision to submit the work for publication.

### Author contributions

TWM-F, Conception and design, Generated transgenic lines, Collected images, and quantified results, Wrote the final document, Critically revised the manuscript and approved the final version for publication; SCP, Conception and design, Generated transgenic lines, Collected images, and quantified results, Critically revised the manuscript and approved the final version for publication; LM, Conception and design, Performed EM experiments, Critically revised the manuscript and approved the final version for publication; CM, Conception and design, Performed oocyte electrophysiology assays, Critically revised the manuscript and approved the final version for publication; MG, Obtained results for unc-55; unc-2; juIs1 strain, Critically revised the manuscript and approved the final version for publication; AB, Obtained results for the unc-55; tax-6; nuIs279 strain, Critically revised the manuscript and approved the final version for publication; SH, SM, Generated the unc-8 (tm5052) mutant allele, Critically revised the manuscript and approved the final version for publication; LB, Designed and interpreted experiments, Critically revised the manuscript and approved the final version for publication; JR, Conception and design, Designed and interpreted experiments and performed in vivo electrophysiological measurements, Critically revised the manuscript and approved the final version for publication; DMM, Conception and design, Designed and interpreted experiments, Wrote the final document, Critically revised the manuscript and approved the final version for publication

### Author ORCIDs

Laura Manning, http://orcid.org/0000-0003-1597-0600
David M Miller III, http://orcid.org/0000-0001-9048-873X

## Additional files

### Supplementary files

• Supplemental file 1. *C. elegans* strains used in this study.

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
