## [Decision Letter]

Thank you for submitting your article "The DEG/ENaC Cation Channel Protein UNC-8 Drives Activity-Dependent Synapse Removal in Remodeling GABAergic Neurons" for consideration by *eLife*. Your article has been reviewed by two peer reviewers, and the evaluation has been overseen by Kang Shen as the Reviewing Editor and Gary Westbrook as the Senior Editor.

The reviewers have discussed the reviews with one another and the Reviewing Editor has drafted this decision to help you prepare a revised submission.

General assessment:

This manuscript addresses the molecular mechanisms that govern a poorly understood, but critical biological process, synapse remodeling. UNC-55 is a transcription factor that prevents the VD class GABAergic motor neurons to adopt the fate of another subclass of GABAergic motor neuron (DD), which undergoes post-embryonic synapse remodeling. Previously, the corresponding author's group performed micro-array experiments to compare the transcriptional profiles of GABAergic motor neurons in the presence and absence of UNC-55, identified a dozen of UNC-55-regulated transcripts (Petersen, JNS, 2011). This manuscript explores the functional involvement of and potential mechanisms through one of these targets, the UNC-8 ENaC sodium channel affect remodeling.

In this paper, the authors used several presynaptic fluorescent markers to quantify the remodeling phenotype of GABAergic motor neurons in wild-type and *unc-55* mutants, and to access the effect of removing UNC-8/ENaC, and its putative pathway components (UNC-2/VGCC, TAX-6/calcineurin, CED-4/Apaf). They confirmed that removing these components, either individually, or in combination with removing UNC-8, led to a similar degree and partial restoration of the ventral presynaptic structures in *unc-55* background. They performed EM analyses to confirm the anatomic presence of these structures, and, by electrophysiology analyses, that these structures (in *unc-55; unc-8*) appeared to lack the ability to release synaptic vesicles. Because a previous study from the group (Petersen, 2011) revealed that IRX-1, a TF, is an UNC-55-target that functions similarly to UNC-8 to promote the remodeling of GABAergic neurons, author probed the genetic relationship between *unc-8* and *irx-1*. They found that removing both in *unc-55* background led to an increased effect on promoting the remodeling. Hence, placing UNC-8 and IRX-1 in two parallel pathways negatively regulated by UNC-55.

Both reviewers found the manuscript contains a lot of data, and two semi-related studies that address two UNC-55-dependent mechanisms that regulate synaptic remodeling. Both reviewers are impressed with the comprehensive EM and electrophysiological analyses. They found that the manuscript is appropriate for publication after the authors address the following questions. As you will see, many of the comments are about writing and the strength of the conclusions. Some comments need to be addressed experimentally.

Essential revisions:

1) Most importantly, both reviewers raised concerns about the interpretation of "synapse elimination" phenotypes in the *unc-8* mutants. One reviewer wrote: "In several places, the manuscript made strong statements along the line that 'UNC-8 drives presynaptic structure removal'. I feel that authors may want to soften these statements and may need additional data to enforce such statement, for following reasons: a) the rescuing effect of removing UNC-8, UNC-2, TAX-6 and CED-4 is not only partial, and appears to reflect morphological, but not functional rescue of *unc-55*'s GABAergic neuron defects. Compared to the effect of removing IRX-1 (Peterson 2011; and this study), the role UNC-8 pathway is less prominent. B) if UNC-2 is to be stated as 'required for remodeling', it would be useful to analyze the remodeling phenotype in *unc-2* single mutants (as shown for *unc-8* in Figure 2).".

Another reviewer wrote: "My main concern of this study is the interpretation of synapse removal. As shown in Figure 2, wt and *unc-8* mutant worms eliminate similar number of synapses between 18 and 30 hours, i.e., about fifty DD synapses (RAB-3 puncta) disappear from the ventral cord. These results seem to indicate that synapse removal remains fully active without UNC-8. Would it be possible that the defects (ventral puncta) in *unc-8* mutants are due to ectopic synapse formation or false deposit of synaptic proteins? Indeed, more RAB-3 puncta exist in *unc-8* mutants prior to remodeling (Figure 2; at 18 hours post-lay).

Similarly, in the case of VD neurons, how do we know that ventral puncta in *unc-55, unc-8* mutants are due to failed synapse elimination, instead of ectopic protein trafficking or synapse formation? Do these fluorescent puncta eventually disappear? Time-course analyses of puncta removal from ventral cord in *unc-55* and *unc-55, unc-8* mutants may be useful to address this issue.

In addition, do these presynaptic puncta have postsynaptic UNC-49::GFP? Because the authors mentioned that UNC-49::GFP clusters remain unchanged, would it be possible that these ventral synapses have lost their postsynaptic partners? If these ventral synapses had postsynaptic receptors prior to remodeling, that could be an indication for failed synapse removal in presynaptic neurons.

Finally, given that no GABA transmission is recovered in *unc-55, unc-8* mutants, the physiological consequence of remaining presynaptic components is unclear.”

2) Presynaptic Marker analyses: Authors accessed the phenotype and genetic interactions between *unc-55* and *unc-8* (and some other suppressors) used several presynaptic markers to conclude that their interaction reflects the change of the entire presynaptic structure, instead of specific effect on individual protein/marker expression or subcellular localization. I like the vigor of this approach. I did notice (e.g. compare Figure 2; Figure 7) that the phenotype, and partial suppression effect of *unc-8* and its putative pathway components, was variable and dependent on the marker used. If I understand the Figure 2 and Figure 3 properly, it seems that the delayed remodeling phenotype of *unc-8* mutants may be more severe with the *Pflp-13* driven SNB-1 than the RAB-3 marker. It would be useful for readers to clarify these points in Discussion.

3) This manuscript may benefit from reorganization and editing. The second part of the story, the analyses between UNC-55 and IRX-1, almost stands alone as a separate study. Having it inserted between that of *unc-8* and *unc-2/tax-6/tom* etc. (Figure 4 and Figure 4—figure supplement 1) cuts the flow of the paper. I suggest moving this part of the study after those defining the genetic pathway of *unc-8*. These sections (the *unc-8* pathway) could be extensively condensed and shortened as it employs very similar assays as those in *unc-8* analyses.

4) This study significantly extends from the 2011 paper. This does not at all reduce my enthusiasm to support it publication, but I think it is important for authors to refer in those places to the previous study so that they can clearly describe and discuss the importance of the extended analyses. For example, the electrophysiology analyses of ventral GABA minis on *unc-55; unc-8* etc. has been extended and complemented with the hyperosmotic responses to pinpoint a potential functional defect in vesicle priming or fusion. This applies similarly to the much more substantial marker analyses on *unc-55; unc-8* mutants.

[Editors' note: further revisions were requested prior to acceptance, as described below.]

Thank you for resubmitting your work entitled "The DEG/ENaC Cation Channel Protein UNC-8 Drives Activity-Dependent Synapse Removal in Remodeling GABAergic Neurons" for further consideration at *eLife*. Your revised article has been favorably evaluated by Gary Westbrook as the Senior editor, a Reviewing editor, and two reviewers.

Both reviewers found that the revision significantly improved the manuscript and found that the manuscript is suitable for publication in *eLife* if these three remaining questions are addressed. We expect a quick turn-around for this last set of questions.

1) Because the authors now demonstrate that in *unc-55; unc-8* double mutants, both DDs and VDs contributed to the ventral residual synaptic marker expression, shouldn't DD in Figure 1's also show both ventral triangles?

2) Why was the electrophysiology data of *unc-8* mutants removed? These data are critical for readers to understand the phenotype of *unc-8* mutants. We think that the Figure 4 in the original manuscript should be shown in the final manuscript.

3) The UNC-49 results were verbally described in the rebuttal letter, but the actual data were not shown in the revised manuscript. To convincingly show synapse elimination/removal, it is important to demonstrate that remaining presynaptic markers have had postsynaptic partners prior to remodeling.

---

## [Author Response]

[…] Both reviewers found the manuscript contains a lot of data, and two semi-related studies that address two UNC-55-dependent mechanisms that regulate synaptic remodeling. Both reviewers are impressed with the comprehensive EM and electrophysiological analyses. They found that the manuscript is appropriate for publication after the authors address the following questions. As you will see, many of the comments are about writing and the strength of the conclusions. Some of the comments need to be addressed experimentally.

We thank reviewers for their appreciation of this work and for many insightful suggestions for improving the manuscript. Of particular note, we have sought to clarify the impression that our results are limited to the ectopic synaptic removal mechanism that occurs in *unc-55* mutants. In fact, our goal is to understand the native synaptic remodeling mechanism (in DD neurons). We have exploited the *unc-55* remodeling phenotype to facilitate experimental analysis and because ample evidence supports the conclusion that the endogenous DD neuron remodeling pathway is activated in *unc-55* mutant VD neurons. For example, multiple components of the DD remodeling mechanism including genes either reported in previous papers (*irx-1, oig-1, hbl-1, ced-4*) or in this work (*unc-8, tax-6/cnb-1*) also promote synaptic removal in *unc-55* mutant VD neurons. Moreover, both DD and VD remodeling programs are activity-dependent. We have emphasized these points in additional text now included in the Introduction and Discussion.

*Essential revisions:*

1) Most importantly, both reviewers raised concerns about the interpretation of "synapse elimination" phenotypes in the unc-8 mutants. One reviewer wrote: "In several places, the manuscript made strong statements along the line that 'UNC-8 drives presynaptic structure removal'. I feel that authors may want to soften these statements and may need additional data to enforce such statement, for following reasons: a) the rescuing effect of removing UNC-8, UNC-2, TAX-6 and CED-4 is not only partial, and appears to reflect morphological, but not functional rescue of unc-55's GABAergic neuron defects. Compared to the effect of removing IRX-1 (Peterson 2011; and this study), the role UNC-8 pathway is less prominent.”

We agree that the role of *unc-8* (and *unc-2, tax-6, ced-4)* in synaptic removal is less prominent that of the transcription factor *irx-1*. We have modified the text to acknowledge this point and to more carefully describe our interpretation of these findings. In particular, we have re-worded statements to the effect that “*unc-8* is required for synaptic removal” to say “*unc-8* promotes synaptic removal.” This alternative description reflects our finding that *unc-8* functions in parallel to other pathways that also promote synaptic removal. In fact, other known components of the remodeling mechanism are also not absolutely required. For example, the apoptotic pathway that removes DD synapses in young larvae is complemented by an additional unknown mechanism that eventually dismantles these connections by the adult stage (Meng et al. 2015). Together, these findings indicate that the synaptic remodeling mechanism is complex and likely involves multiple pathways. We have included the results of an additional experiment that validates the role of *unc-8* in synaptic removal. We interpret our genetic results to indicate that wild-type *unc-8* is required for the removal of a significant fraction of presynaptic domains in remodeling GABAergic neurons. To test the hypothesis that *unc-8* function is also sufficient to drive synapse removal, we have used a transgenic strategy to over-express *unc-8* in ventral cord GABAergic neurons. In this experiment, we observed that ventral synapses are substantially reduced in VD neurons that express ectopic *unc-8*, whereas neighboring VD neurons that do not express *unc-8* retain the wild-type complement of ventral presynaptic domains. This result supports the model that UNC-8 drives presynaptic removal. This experiment has been included in Figure 3—figure supplement 2.

“b) if UNC-2 is to be stated as 'required for remodeling', it would be useful to analyze the remodeling phenotype in unc-2 single mutants (as shown for unc-8 in Figure 2).".

We thank the reviewer for this suggestion. Our genetic results have shown that *unc-2* promotes synaptic removal in the *unc-55* remodeling program. To directly test the proposal that *unc-2* is also involved in DD remodeling, we used the DD::mCherry::RAB-3 marker to monitor the DD presynaptic domain in *unc-2* mutants. The results of this experiment were inconclusive, however. Although the removal of DD presynaptic domains was delayed in an *unc-2* mutant in comparison to wild type, we also noted that *unc-2* mutants develop more slowly as well. In the future, we could potentially obviate this problem by using cell-specific RNAi to target *unc-2* specifically in GABA neurons.

Another reviewer wrote: "My main concern of this study is the interpretation of synapse removal. As shown in Figure 2, wt and unc-8 mutant worms eliminate similar number of synapses between 18 and 30 hours, i.e., about fifty DD synapses (RAB-3 puncta) disappear from the ventral cord. These results seem to indicate that synapse removal remains fully active without UNC-8. Would it be possible that the defects (ventral puncta) in unc-8 mutants are due to ectopic synapse formation or false deposit of synaptic proteins? Indeed, more RAB-3 puncta exist in unc-8 mutants prior to remodeling (Figure 2; at 18 hours post-lay).”

Thank you, this is an insightful observation. We have now quantified the number of synapses in wild-type and *unc-8* animals prior to the 18-hour time point. This experiment shows that wild-type and *unc-8* animals display similar numbers of ventral puncta at this stage. The higher number of puncta observed in *unc-8* animals at later time points is consistent with the proposal that removal of ventral DD synapses is delayed in *unc-8* mutants in comparison to wild type. These new data are now included in Figure 2.

“Similarly, in the case of VD neurons, how do we know that ventral puncta in unc-55, unc-8 mutants are due to failed synapse elimination, instead of ectopic protein trafficking or synapse formation? Do these fluorescent puncta eventually disappear? Time-course analyses of puncta removal from ventral cord in unc-55 and unc-55, unc-8 mutants may be useful to address this issue.”

Thank you for this observation and experimental suggestion. We performed time-course analysis in both *unc-55* and *unc-55; unc-8* animals (See Figure 3—figure supplement 1). At early time points (12, 18, 25, and 36 hours post lay) *unc-55* and *unc-55; unc-8* animals show similar numbers of ventral synapses. At later stages (> 48 hour post lay), as previously reported in Petersen et al. (2011), the number of ventral GABA synapses declines in *unc-55* mutants. In contrast, the number of ventral synapses in the *unc-55; unc-8* animals remains steady (through 96 hours post lay) and does not decline. These results are consistent with the idea that the residual ventral synapses that we detected by fluorescence and electron microscopy in *unc-55; unc-8* adults are likely to correspond to GABA synapses that are not properly removed as opposed to new synapses which might otherwise appear in this experimental approach as a spike in ventral puncta during later development of *unc-55; unc-8* animals. Additionally, it is unlikely that the GABA synapses we see by EM in the ventral nerve cords of *unc-55; unc-8* animals are due to ectopic protein trafficking, as they include organized presynaptic structures with active zones, synaptic vesicles, dense core vesicles, and juxtaposed postsynaptic muscle arms.

“In addition, do these presynaptic puncta have postsynaptic UNC-49::GFP? Because the authors mentioned that UNC-49::GFP clusters remain unchanged, would it be possible that these ventral synapses have lost their postsynaptic partners? If these ventral synapses had postsynaptic receptors prior to remodeling, that could be an indication for failed synapse removal in presynaptic neurons.

Finally, given that no GABA transmission is recovered in unc-55, unc-8 mutants, the physiological consequence of remaining presynaptic components is unclear.”

Thank you for this suggestion. We have performed experiments to test whether the ventral presynaptic puncta in *unc-55; unc-8* GABA neurons co-localize with postsynaptic UNC-49::GFP. These results (described below) are not included in the revised version of this paper, however, because we have also removed electrophysiological data indicating that residual ventral synapses in *unc-55; unc-8* animals are not functional and will publish these findings in a separate paper separate manuscript featuring the parallel roles of *unc-8* and *irx-1* during synaptic remodeling. We captured images of wild-type, *unc-8*, and *unc-55; unc-8* animals co-expressing a presynaptic GABA marker (*punc-25::mCherry::RAB-3*) and a postsynaptic GABA receptor marker (*punc-49::UNC-49::GFP*) and analyzed these images for co-localization using ImageJ software. Although, co-localization of mCherry::RAB-3 and UNC-49::GFP in *unc-55; unc-*8 animals (r^2^= 0.53) is decreased in comparison to wild-type and *unc-8* animals (r^2^ = 0.80 and 0.77, respectively), we attribute this difference to the observation that there is less punc-25::mCherry::RAB-3 ventral puncta in *unc-55; unc-8* animals compared to wild type. This observation suggests that these synapses have not lost their postsynaptic partners.

Reviewers are correct to note that GABA synaptic function is not restored in *unc-55; unc-8* animals. We agree that these apparently non-functional presynaptic GABAergic domains are not physiologically competent. However, the existence of these presynaptic structures in an *unc-55; unc-8* mutant *whether functional or not* provides clear evidence that wild-type UNC-8 promotes the removal of presynaptic components including proteins that are required for neurotransmitter release (e.g., RAB-3, SNB-1/V-SNARE, UNC-57/endophilin, SYD-2/Liprin). The non-functionality of the presynaptic regions that are *not* removed in *unc-8* mutants suggests that an additional parallel pathway, which is still active in an *unc-8* mutant, must account for this deficit. At the recommendation of the reviewers (see below) who felt that our analysis of a second parallel acting remodeling pathway detracted from the main point of this work (i.e., UNC-8 promotes synaptic remodeling) these data have been removed from the paper and will be featured in a separate manuscript that documents the parallel roles of the *unc-8* and *irx-1*-dependent remodeling pathways.

2) Presynaptic Marker analyses: Authors accessed the phenotype and genetic interactions between unc-55 and unc-8 (and some other suppressors) used several presynaptic markers to conclude that their interaction reflects the change of the entire presynaptic structure, instead of specific effect on individual protein/marker expression or subcellular localization. I like the vigor of this approach. I did notice (e.g. compare Figure 2; Figure 7) that the phenotype, and partial suppression effect of unc-8 and its putative pathway components, was variable and dependent on the marker used. If I understand the Figure 2 and Figure 3 properly, it seems that the delayed remodeling phenotype of unc-8 mutants may be more severe with the Pflp-13 driven SNB-1 than the RAB-3 marker. It would be useful for readers to clarify these points in Discussion.

Thank you for this comment. We apologize for the confusion and have sought to clarify this point by noting that we used two different paradigms to monitor synapse removal (e.g., the native DD remodeling program and ectopic remodeling in *unc-55* mutant VD neurons). The data in Figure 2 were collected using a DD-specific promoter to monitor the removal defect specifically in DD neurons. We have sought to clarify this point by labeling all the experiments where we used the DD-specific marker as “*DD::SNB-1::GFP* or *DD::GFP::RAB-3*” rather than using the less descriptive nomenclature “*pflp-13::.*.”. Additionally, in Figure 3–Figure 8 we have labeled the DD and VD markers as “GABA::SNB-1::GFP”, etc. rather than with the *C. elegans* gene names “*punc-25/pttr-39*”. Finally, these figures now include additional schematics depicting the type of remodeling GABA neuron (DD or VD) to help readers recognize each of these specific experimental paradigms. We have also modified the text to clarify this point.

3) This manuscript may benefit from reorganization and editing. The second part of the story, the analyses between UNC-55 and IRX-1, almost stands alone as a separate study. Having it inserted between that of unc-8 and unc-2/tax-6/tom etc. (Figure 4 and Figure 4—figure supplement 1) cuts the flow of the paper. I suggest to move this part of the study after those defining the genetic pathway of unc-8. These sections (the unc-8 pathway) could be extensively condensed and shortened as it employs very similar assays as those in unc-8 analyses.

Thank you for this suggestion. We agree that our analysis of *irx-1* detracts from the main thrust of the paper and that it could stand alone as a separate study. Accordingly, we have removed these data from the paper for a separate manuscript.

*4) This study significantly extends from the 2011 paper. This does not at all reduce my enthusiasm to support it publication, but I think it is important for authors to refer in those places to the previous study so that they can clearly describe and discuss the importance of the extended analyses. For example, the electrophysiology analyses of ventral GABA minis on unc-55; unc-8 etc. has been extended and complemented with the hyperosmotic responses to pinpoint a potential functional defect in vesicle priming or fusion. This applies similarly to the much more substantial marker analyses on unc-55; unc-8 mutants.*

Thank you for noting that this work significantly extends our previous findings. Our earlier paper (Petersen et al., 2011) provided the initial evidence of a role for UNC-8 in synaptic remodeling by detecting *unc-8* as an *unc-55*-regulated transcript and by demonstrating that RNAi knockdown of *unc-8* suppresses *unc-55* (i.e., retards synaptic removal). We have summarized these results here and emphasized additional experiments in this paper that extend our original findings. For example, the first Results section uses an *unc-8* promoter-GFP reporter gene to confirm the prediction that *unc-8* is negatively regulated by *unc-55* in VD neurons. Next, we report experiments with a new *unc-8* deletion allele that confirm the initial RNAi results and which show, for the first time, that unc-8 also functions in DD neurons to promote synapse removal in the native GABA neuron remodeling pathway. Electrophysiological data (e.g., “…hyperosmotic responses to pinpoint a potential functional defect in vesicle priming…”) have been removed from this paper for a separate manuscript that will test a specific molecular model to explain this effect.

[Editors' note: further revisions were requested prior to acceptance, as described below.]

*Both reviewers found that the revision significantly improved the manuscript and found that the manuscript is suitable for publication in eLife if these three remaining questions are addressed. We expect a quick turn-around for this last set of questions.*

1) Because the authors now demonstrate that in unc-55; unc-8 double mutants, both DDs and VDs contributed to the ventral residual synaptic marker expression, shouldn't DD in Figure 1's also show both ventral triangles?

Yes, we agree that this work demonstrates a role for *unc-8* in the removal of both DD and VD ventral synapses, and the schematic in Figure 1 has been modified to reflect this conclusion.

2) Why was the electrophysiology data of unc-8 mutants removed? These data are critical for readers to understand the phenotype of unc-8 mutants. We think that the Figure 4 in the original manuscript should be shown in the final manuscript.

We have restored the original electrophysiology data (Figure 4) and controls (Figure 4—figure supplement 1, panels E and F). We are happy with this arrangement but note that inclusion of the electrophysiology data showing that *unc-55; unc-8* synapses are non-functional also required us to suggest that an additional pathway, acting in parallel, likely accounts for this effect (see subsection “UNC-8 channel activity is required for synapse removal”).

*3) The UNC-49 results were verbally described in the rebuttal letter, but the actual data were not shown in the revised manuscript. To convincingly show synapse elimination/removal, it is important to demonstrate that remaining presynaptic markers have had postsynaptic partners prior to remodeling.*

These data are now included (Figure 4—figure supplement 3). We show that pre- and postsynaptic compartments of wild-type and *unc-8* animals (as well as *unc-55* and *unc-55; unc-8* animals) co-localize in larval animals prior to remodeling and are also juxtaposed after remodeling.